# Changes in the genetic requirements for microbial interactions with increasing community complexity

Manon Morin[1], Emily C Pierce[1], Rachel J Dutton[1,2]*

[1]Division of Biological Sciences, University of California, San Diego, La Jolla, United States; [2]Center for Microbiome Innovation, Jacobs School of Engineering, University of California San Diego, La Jolla, United States

**Abstract** Microbial community structure and function rely on complex interactions whose underlying molecular mechanisms are poorly understood. To investigate these interactions in a simple microbiome, we introduced *E. coli* into an experimental community based on a cheese rind and identified the differences in *E. coli*'s genetic requirements for growth in interactive and non-interactive contexts using Random Barcode Transposon Sequencing (RB-TnSeq) and RNASeq. Genetic requirements varied among pairwise growth conditions and between pairwise and community conditions. Our analysis points to mechanisms by which growth conditions change as a result of increasing community complexity and suggests that growth within a community relies on a combination of pairwise and higher-order interactions. Our work provides a framework for using the model organism *E. coli* as a readout to investigate microbial interactions regardless of the genetic tractability of members of the studied ecosystem.
DOI: https://doi.org/10.7554/eLife.37072.001

## Introduction

Microorganisms rarely grow as single isolated species but rather as part of diverse microbial communities. In these communities, bacteria, archaea, protists, viruses and fungi can coexist and perform complex functions impacting biogeochemical cycles and human health (*Falkowski et al., 2008*; *Flint et al., 2012*). Deciphering microbial growth principles within a community is challenging due to the intricate interactions between microorganisms, and between microorganisms and their environment. While interest in microbial communities has dramatically increased, our understanding of microbial interactions within communities is lagging significantly behind our ability to describe the composition of a given community.

Approaches relying on 16S rDNA sequencing analyses of microbial communities can be used to reconstruct ecosystem networks and detect patterns of co-occurrence to infer general interactions such as competition, mutualism and commensalism (*Faust and Raes, 2012*). However, the molecular mechanisms underlying these interactions remain largely uncharacterized. Further, the way in which these interactions are organized within a community, such as whether they consist of predominantly pairwise or higher-order interactions, is even less clear. A more precise understanding of microbial interactions, their underlying mechanisms, and how these interactions are structured within a community, are all necessary to elucidate the principles by which a community is shaped. In this study, we combine genome-scale genetic and transcriptomic approaches within an experimentally tractable model microbial community to begin to address these questions.

Genome-scale approaches, such as transposon mutagenesis coupled to next-generation sequencing (TnSeq approaches) have been successfully used to quantify the contribution and thus the importance of individual genes to a given phenotype (*van Opijnen and Camilli, 2013*). These techniques

*For correspondence:
rjdutton@ucsd.edu

Competing interests: The authors declare that no competing interests exist.

**eLife digest** Microorganisms live almost everywhere on Earth. Whether it is rainforest soil or human skin, each environment hosts a unique community of microbes, referred to as its microbiome. There can be upwards of hundreds of species in a single microbiome, and these species can interact in a variety of ways; some cooperate, others compete, and some can kill other species. Deciphering the nature of these interactions is crucial to knowing how microbiomes work, and how they might be manipulated, for example, to improve human health. Yet studies into these interactions have proven difficult, not least because most of the species involved are difficult to grow in controlled experiments.

One environment that is home to a rich community of microbes is the outer surface of cheese, known as the cheese rind. The cheese rind microbiome is a useful system for laboratory experiments, because it is relatively easy to replicate and its microbes can be grown on their own or in combinations with others.

To explore the nature of interactions in microbiomes, Morin et al. have now grown a large collection of *E. coli* mutants as members of simplified microbiomes based on the cheese rind. The mutant bacteria were grown on cheese either alone, paired with one other species, or alongside a community of three species. The aim was to see which mutants grew poorly when other species were present, thus allowing Morin et al. to identify specific genes that are important for interactions within the experimental microbiomes.

Even in these simplified microbiomes, the microbes interacted in a variety of ways. Some microbes competed with *E. coli* for elements like iron and nitrogen; others cooperated by sharing the building blocks needed to make larger molecules. Many of the interactions that happened when *E. coli* was paired with one species were not seen when more species were added to the community. Similarly, some interactions were only seen when *E. coli* was grown alongside a community of microbes, and not when it was paired with any of the three species on their own.

These findings show that complex interactions are present even in a simplified microbiome. This experimental approach can now be applied to other microbiomes that can be grown in the laboratory to examine whether the patterns of interactions seen are generalizable or specific to the cheese rind system.

DOI: https://doi.org/10.7554/eLife.37072.002

use a pooled library of transposon insertion mutants whose frequency is measured to identify genes important for growth in a given condition. Recently, the generation and introduction of unique random barcodes into transposon mutant libraries made this approach more high-throughput, enabling screens of important genes across hundreds of conditions and for numerous genetically tractable microorganisms (*Wetmore et al., 2015*; *Price et al., 2018*).

To investigate the genetic basis of microbial interactions, we have adapted this approach to identify and compare genetic requirements in single-species (non-interactive) and multi-species (interactive) conditions. We used a large and diverse transposon library previously generated in the genetically-tractable model bacterium *E. coli* K12 (*Wetmore et al., 2015*) to characterize the genetic requirements of interactions within a model community based on the rind of cheese (*Wolfe et al., 2014*). The fact that the *E. coli* genome has undergone extensive characterization can help more effectively interpret the genetic requirements introduced by growth within communities. Although *E. coli* K12 is not a typical endogenous species of this particular microbiome, non-pathogenic *E. coli* strains can be found in raw milk and raw-milk cheese (*Trmčić et al., 2016*). Shiga-toxin-producing *E. coli* 0157:H7 and non-0157 pathogenic *E. coli* species are common invaders of the cheese environment and can survive during cheese making causing mild to life-threatening symptoms after ingestion (*Coia et al., 2001*; *Montet et al., 2009*; *Frank et al., 1977*).

Using the *E. coli* transposon library, we (i) identified the set of genes important for growth alone in the cheese environment, (ii) identified the set of genes important for growth in pairwise conditions with each individual community member and (iii) identified the set of genes important for growth in the presence of the complete community. Characterization of the functions or pathways associated with growth in interactive versus non-interactive conditions were then used to infer the biological

processes involved in interactions within the model microbiome. Additionally, we compared the set of genes important for growth in pairwise conditions with the ones important for growth in a community to investigate how microbial interactions change depending on the complexity of the interactive context. We also performed a similar RB-TnSeq analysis during non-interactive and interactive conditions using a transposon library we generated in the cheese-endogenous species *Pseudomonas psychrophila* JB418. Finally, we measured changes in the transcriptional profile of *E. coli* during growth alone, growth in pairwise conditions, and within the community using RNAseq as a complementary approach to RB-TnSeq in defining microbial interactions. This analysis revealed a deep reorganization of gene expression whenever *E. coli* is in the presence of other species.

This work revealed numerous interactions between species, such as metabolic competition for iron and nitrogen, as well as cross-feeding from fungal partners for certain amino acids. Our analysis showed that most of the metabolic interactions (competition and cross-feeding) observed in pairwise conditions are maintained and amplified by the addition of all partners in the community context. However, around half of the genetic requirements observed in pairwise conditions were no longer apparent in the community, suggesting that higher-order interactions emerge in a community.

## Results

### Identification of the basic genetic requirements for growth of *E. coli* in isolation

We used the *E. coli* Keio_ML9 RB-TnSeq library from *Wetmore et al., 2015*, containing a pool of 152,018 different insertion mutants (with a median of 16 insertion mutants per gene; covering 3728 of 4146 protein-coding genes), each associated with a unique 20 nucleotide barcode. This library was originally generated in and maintained on lysogeny broth medium (LB) and was used previously to identify genes required for growth across a variety of conditions (*Wetmore et al., 2015*; *Price et al., 2018*). To determine genes important for growth on our cheese-based medium, we grew the pooled library by itself on sterile cheese curd agar plates (CCA: 10% freeze-dried fresh cheese, 3% NaCl, 0.5% xanthan gum, 1.7% agar), the same medium used in all further experiments and used previously to demonstrate that cheese communities could be successfully reconstructed *in vitro* (*Wolfe et al., 2014*). As the library is composed of multiple insertion mutants for a gene, we expect the individual insertion mutants to be evenly distributed in the experimental environment, minimizing the effect on any individual insertion mutant due to stochastic processes such as genetic drift or localized effects related to spatial structure (*Hallatschek et al., 2007*). During growth, we expect the library to modify the environment by taking up nutrients and excreting molecules (waste products, enzymes, etc). Consequently, we expect that some genetic requirements will change during growth. Thus, to provide a comprehensive overview of the genetic requirements for growth, we grew the pooled library on CCA and collected samples after 1, 2 and 3 days. For each time point, we harvested the library from the surface of the cheese plate, extracted genomic DNA, used PCR to amplify the barcoded regions of the transposons, and then sequenced these products to measure the abundance (*i.e.* the number of sequencing reads associated with each barcode) of each transposon mutant over time (see Materials and Methods).

The fitness of each insertion mutant was calculated as the $\log_2$ of the ratio of its abundance at a given timepoint compared to its abundance at T0 (the inoculum). We calculated the raw fitness of a gene as the weighted average of the fitness of all insertion mutants of that gene. Gene fitness values were then normalized. First, fitness values are corrected to account for changes in copy number along the chromosome as insertions near the replication fork are expected to have higher copies in dividing cells. Then, fitness values were normalized based on the assumption that disruption of most of the genes leads to little or no fitness effect (see Materials and Methods and (*Wetmore et al., 2015*) for details). Consequently, most of the fitness values are expected to be close to 0, indicating that disruption of these genes leads to no particular growth modification compared to the rest of the library. Negative gene fitness values, however, identify mutants that are growing slower than the rest of the library and therefore, genes that are of particular importance for growth in the studied condition. A t-score, calculated as a moderated t-statistic, is determined for each gene fitness value to assess if the fitness value is reliably different from 0 (see Materials and Methods and

(*Wetmore et al., 2015*) for details). The RB-TnSeq pipeline from experimental set-up to gene fitness calculation is summarized in *Figure 1—figure supplement 1*.

At each timepoint, we were able to calculate the fitness and a corresponding t-score for 3298 protein-coding genes (*Figure 1A*, *Figure 1—source data 1*). Because we were interested in genes with a strong fitness defect (significant negative fitness values), we first removed genes with an absolute t-score <= 3. This t-score threshold was set to identify strong negative fitness values while minimizing potential false positives (false discovery rate of 0.2%). The t-score assesses how reliably a fitness value is different from 0. In each condition, most genes have no detectable fitness effect, and thus have a fitness value close to 0. Thus, in our dataset, most of the genes below this t-score also have a fitness value close to 0. On average, 97% of the fitness values were associated with a t-score that falls below our threshold. Within the fitness values that pass the t-score threshold, we then removed genes associated with positive fitness values. Thus, we only retained the genes whose deletion leads to a consistent growth defect for *E. coli* on CCA compared to the rest of the library. This filtering process revealed 160 genes that were important for *E. coli* growth alone on CCA (*Figure 1A*).

To identify the functions associated with these 160 genes, we mapped them to the KEGG BRITE database (*Figure 1B*). 84 genes were assigned to KEGG modules and 64 of them were associated with *E. coli* metabolism. Within these metabolic genes, we found 28 genes associated with amino acid metabolism, specifically the biosynthesis of all amino acids except for proline, lysine and histidine. Quantification of free amino acids in our medium highlighted very low concentrations of all amino acids (*Figure 1—figure supplement 2*) suggesting that a limited supply of free amino acids leads to a genetic requirement for amino acid biosynthesis. This is supported by the observation that both *spoT* and *relA*, regulators of the stringent response which can be triggered by amino acid starvation (*Cashel et al., 1996*), are also associated with a negative fitness value. Additionally, we observed the importance of the regulator *gcvR*, that inhibits catabolism of glycine into C1 metabolism. In fact, GcvR inhibits the glycine cleavage complex, a multienzyme complex that oxidizes glycine (*Ghrist and Stauffer, 1995*; *Ghrist et al., 2001*). Furthermore, mutants of the glycine cleavage complex displayed a significant positive fitness, suggesting that absence of glycine utilization through C1 metabolism is beneficial in our amino acid-deficient environment. Altogether, this observation also underlines that amino acids are limiting in the environment and that their biosynthesis and utilization control is important for growth. 19 of the 160 genes were associated with energy metabolism and, more specifically, with sulfur assimilation (n = 7 genes) and respiration (n = 8 genes). Here, we deduce that importance of sulfur assimilation is directly caused by the lack of the amino acids cysteine and methionine, which are the major pools of sulfur-containing compounds in the cell. As a non-endogenous species, *E. coli* might not possess the adequate peptidases or proteases to degrade and use the highly available protein casein. Identification of two of the three genes of the Leloir pathway (*galE* and *galT*), involved in the uptake and conversion of galactose into glucose, suggests that galactose might be a crucial nutrient for *E. coli* growth on CCA.

Eight genes mapped to membrane transport and were associated with two specific pathways: ferric-enterobactin transport and glycine-betaine transport. Ferric-enterobactin transport allows the cells to scavenge iron in a low-iron environment (*Raymond et al., 2003*; *Hider and Kong, 2010*). Iron is an essential micronutrient and cheese is known to be iron-limited (*Albar et al., 2014*). Glycine betaine is used by cells as an osmoprotectant in high osmolarity environments. During cheese curd processing, high concentrations of NaCl are added (*Guinee, 2004*), and our CCA medium contains 3% NaCl to mimic these conditions. The importance for *E. coli* to maintain its cell osmolarity is also suggested by the requirement of genes responsible for the transport of the ions sodium, potassium and zinc.

In our experiment, the fact that all of the mutants are pooled together limits our ability to identify genes whose phenotypes can be complemented by common goods (molecules released in the environment) produced by neighboring cells. For example, given that iron is limiting in cheese, we expect that enterobactin biosynthesis is an important pathway for growth in this environment. However, no genes from the enterobactin biosynthesis pathway (*entCEBAH*, *entD* and *entF*) had a significant negative fitness value (average fitness of the enterobactin biosynthesis pathway: 0.1), while individual growth of these enterobactin biosynthesis mutants from the KEIO collection was limited on CCA compared to a rich, non-iron-limited medium (*Figure 1—figure supplement 3*).

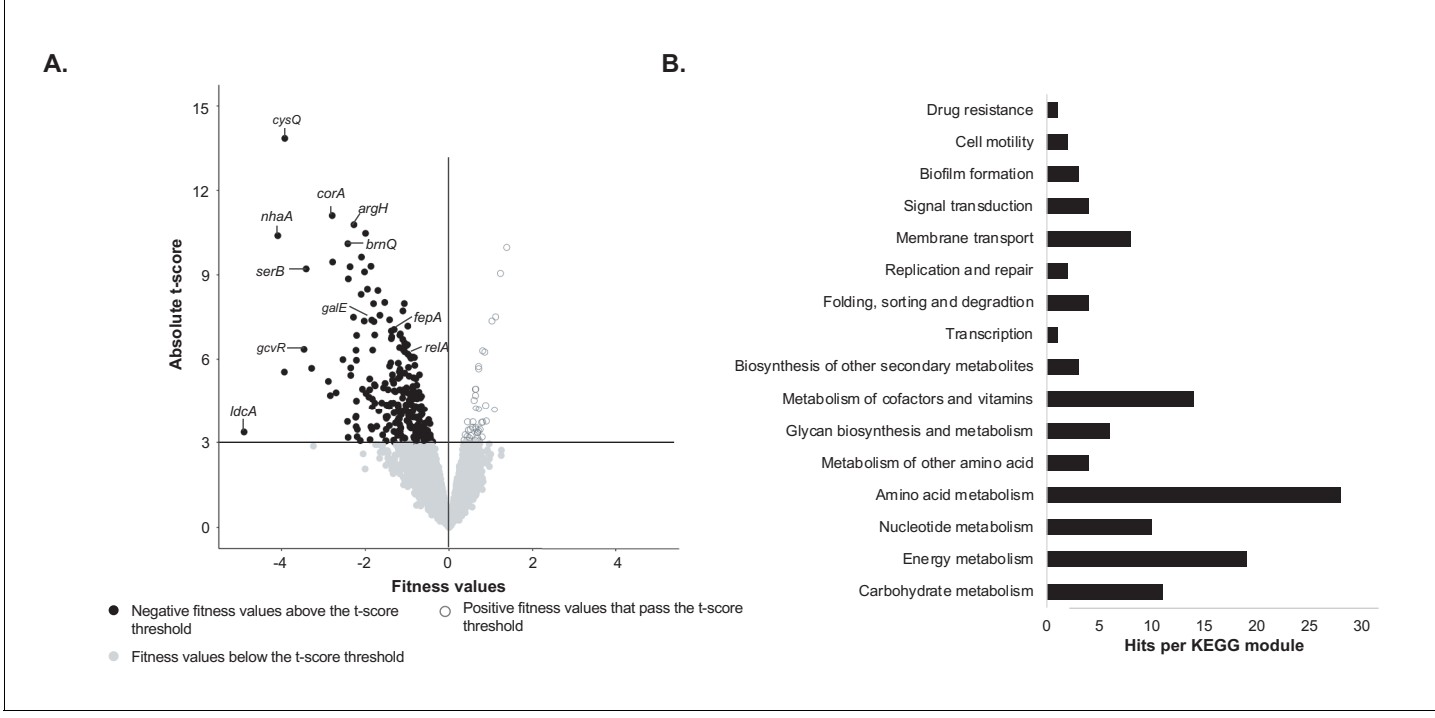

**Figure 1.** Identification of genes important for growth of *E. coli* alone on cheese curd agar (*Figure 1—source data 1*). (**A**) The pooled *E. coli* RB-TnSeq library Keio_ML9 (*Wetmore et al., 2015*) was grown alone on cheese curd agar (CCA). Gene fitness values were calculated for 3298 genes at days 1, 2, and 3 along with a t-score, which assesses how reliably the fitness value differs from 0. The fitness values obtained at the three timepoints are displayed on a single volcano plot (3 points per gene). A t-score threshold of absolute(t-score)≥3 was used to identify genes with strong fitness effects. 97% of the genes fell below this threshold and have no strong and significant fitness effect. Black dots represent genes with strong negative fitness effects. Altogether, they represent 160 different genes that are associated with a significant fitness value for at least one timepoint. (**B**) These 160 genes were mapped to the KEGG BRITE Database for functional analysis and identification of required functions for *E. coli* growth on CCA. 84 of the 160 genes had hits when mapped to the KEGG BRITE database.

DOI: https://doi.org/10.7554/eLife.37072.003

The following source data and figure supplements are available for figure 1:

**Source data 1.** RB-TnSeq analysis of *E. coli's* growth alone on 10% cheese curd agar, pH7.
DOI: https://doi.org/10.7554/eLife.37072.012

**Figure supplement 1.** Pipeline of RB-TnSeq experiment using the *E. coli* Keio M9 library: from experimental set-up to normalized gene fitness and t-score calculation.
DOI: https://doi.org/10.7554/eLife.37072.004

**Figure supplement 2.** Quantification of free amino acids in CCA.
DOI: https://doi.org/10.7554/eLife.37072.005

**Figure supplement 3.** Comparison of individual growth of enterobactin biosynthesis mutants on LB and CCA.
DOI: https://doi.org/10.7554/eLife.37072.006

**Figure supplement 4.** RB-TnSeq experiments using the P. psychrophila JB418 library.
DOI: https://doi.org/10.7554/eLife.37072.007

**Figure supplement 4—source data 1.** RB-TnSeq analysis of *P. psychrophila's* growth alone, in pairwise conditions and with the community on 10% cheese curd agar, pH7.
DOI: https://doi.org/10.7554/eLife.37072.008

**Figure supplement 5.** Competitive assays of 25 mutants of the Keio collection (*Baba et al., 2006*).
DOI: https://doi.org/10.7554/eLife.37072.009

**Figure supplement 6.** Map of the JB418_ECP1 transposon library generated in *P. psychrophila* JB418.
DOI: https://doi.org/10.7554/eLife.37072.010

**Figure supplement 7.** Quality assessment of all RB-TnSeq experiments.
DOI: https://doi.org/10.7554/eLife.37072.011

In summary, functions of major importance for *E. coli* to grow alone in our experimental environment involved (i) response to low iron availability, (ii) response to osmotic stress and (iii) response to limited available nutrients (specifically free amino acids). These required functions are consistent with recently published results on the requirements of the mammary pathogenic *E. coli* (MPEC) during growth in milk (*Olson et al., 2018*) except for resistance to osmotic stress which does not occur in milk. We also generated an RB-TnSeq library in the bloomy rind cheese endogenous species *P. psychrophila* JB418 and found comparable requirements for growth alone on cheese (*Figure 1—figure supplement 4*).

To validate the results obtained with the RB-TnSeq library, we measured the fitness of individual knockout mutants from the *E. coli* Keio collection (*Baba et al., 2006*). We tested 25 knockout mutants corresponding to genes with a strong growth defect observed after one day of growth. We carried out competitive assays between each knockout mutant and the wild-type strain on CCA. We calculated each knockout mutant fitness as the log2 of the fold change of its abundance after one day of growth. A z-score was also calculated to assess the confidence of that fitness. 21 of 25 knockout mutants displayed a fitness value lower than 0 with at least 95% confidence (*Figure 1—figure supplement 5*). The remaining four mutant strains (*brnQ*, *cysK*, *serA* and *trxA*) were associated with high fitness value variability across replicate experiments and had a lower z-score. Altogether, this supports the reliability and validity of RB-TnSeq results.

## Identification of *E. coli* genetic requirements for growth in pairwise conditions

The growth of the *E. coli* library alone allowed us to determine the baseline set of genes required for optimal growth in the model cheese environment. We next wanted to identify genes with negative fitness during growth when another species is present. First, we analyzed the growth of *E. coli* and the partner species. We grew *E. coli* for 3 days on CCA in the presence of either *H. alvei*, *G. candidum* or *P. camemberti*. In addition to belonging to distinct domains or phyla, these three partners are the typical members of a bloomy rind cheese community (such as Brie or Camembert). The presence of *E. coli* did not influence the growth of any partner species (*Figure 2—figure supplement 1*). However, *E. coli*'s growth was reduced in the presence of each partner after three days of growth (*Figure 2A*).

We then determined the genes associated with negative fitness during *E. coli* growth in each pairwise condition using RB-TnSeq (*i.e.* genes whose fitness value is negative and associated with an absolute t-score greater than three in the pairwise condition) (*Figure 2B*, *Figure 2—source data 1*). As performed above, barcode frequencies were compared between T0 and after growth with each partner (at days 1, 2 and 3). As our goal is to compare genetic requirements for growth in interactive and non-interactive conditions rather than to examine changes in requirements over time, we grouped genes with a significant negative fitness for at least one timepoint as a single set of genes for each pairwise condition. We identified 145 genes with negative fitness values in *E. coli* for growth with *H. alvei*, 142 genes for growth with *G. candidum* and 131 genes for growth with *P. camemberti*. Altogether they constitute a set of 153 genes that are required for optimal growth in at least one pairwise culture.

Comparison of genes with negative fitness identified when *E. coli* is grown alone with the genes identified when *E. coli* is grown in pairwise conditions is expected to highlight differences brought about by the presence of another species (*Figure 2C*). Consistent presence of multiple genes of the same pathway within only one of these sets of genes associated with negative fitness is likely to point out a pathway specifically important in one condition. Thus, we can infer possible interactions based on the different relevant pathways between interactive and non-interactive growth conditions. Altogether, the 153 genes with a negative fitness in pairwise conditions and the 160 genes for *E. coli* growth alone represent 235 unique genes (*Figure 2C*). These can be divided into three groups of genes: (i) conserved negative fitness: genes with negative fitness in both growth alone and in all pairwise conditions (n = 78), (ii) pairwise-alleviated negative fitness: any gene found to have a negative fitness during *E. coli* growth alone that was not associated with a negative fitness in at least one of the three pairwise cultures (n = 82), and (iii) pairwise-induced negative fitness: any gene with negative fitness in the presence of at least one of the partners but not associated with a negative fitness during growth alone (n = 75) (*Figure 2C and D* and *Figure 2—figure supplement 2*). We further

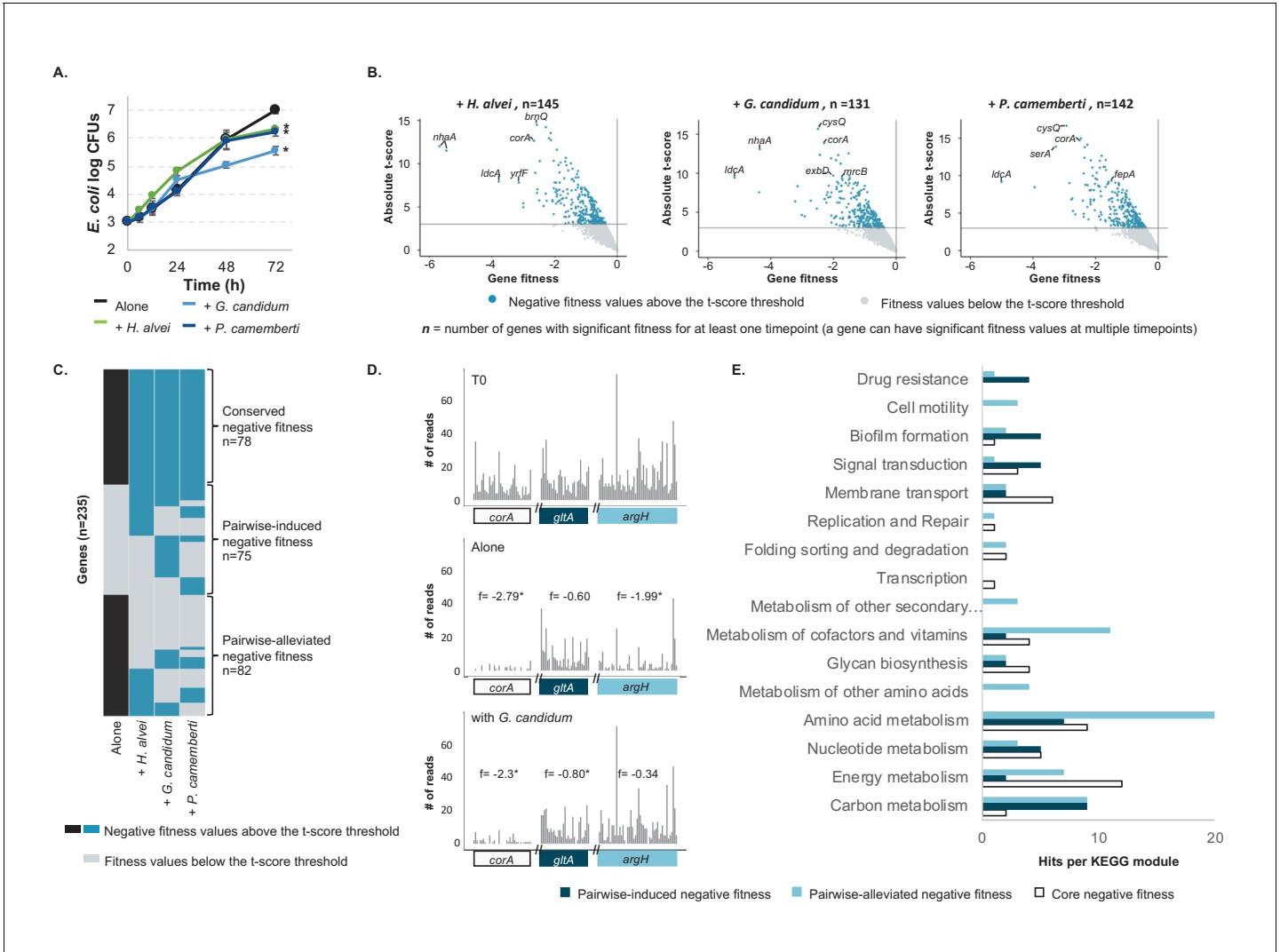

**Figure 2.** *E. coli* genes with negative fitness during growth in pairwise conditions (*Figure 2—source data 1*). (A) We grew *E. coli* in pairwise conditions on CCA with either *H. alvei*, *G. candidum* or *P. camemberti*. Asterisks indicate significant differences in growth of *E. coli* as compared to growth alone at day 3 (Dunnett's test, adjusted p-value≤5%) (B) Using the *E. coli* RB-TnSeq library, we identified genes with negative fitness in each pairwise condition at three timepoints (days 1, 2, 3). Each volcano plot shows fitness values of all 3298 genes at all timepoints (three points per gene). We identified 145 genes with a negative fitness in the presence of *H. alvei* in at least one timepoint, 131 genes in pairwise culture with *G. candidum* and 142 genes in pairwise culture with *P. camemberti*. Altogether, they constitute 153 genes with negative fitness in pairwise conditions. (C) Comparing these genes (dark blue) to the 160 genes with a negative fitness during *E. coli* growth alone (black), we obtained a total of 235 unique genes and identified 78 genes that have a negative fitness both during growth alone and all pairwise conditions (conserved negative fitness), 75 genes that have a negative fitness in at least one pairwise condition but not alone (pairwise-induced negative fitness) and 82 genes with a negative fitness in growth alone but not in at least one pairwise condition (pairwise-alleviated negative fitness). (D) We selected a gene to illustrate conserved negative fitness (*corA*, 37 insertion mutants), pairwise-induced negative fitness (*lpoB*, 31 insertion mutants), and to illustrate pairwise-alleviated negative fitness (*argH*, 50 insertion mutants). For each gene, we display the number of sequencing reads for associated insertion mutants in the T0 sample, in growth alone day 3 and growth with *G. candidum* day 3. These sequencing reads are the raw data accounting for mutant abundance and used for fitness calculation (f represents each gene's fitness value). While reads are not rarefied in the fitness calculation pipeline, we used rarefied reads for the purpose of the figure. Asterisks indicate genes with significant fitness values (consistent decrease in the number of reads per insertion mutant in the condition compared to T0). (E) We mapped the genes associated with conserved, pairwise-induced, and pairwise-alleviated negative fitness to the KEGG BRITE database. 41/75, 45/82 and 33/77 genes had hits.

DOI: https://doi.org/10.7554/eLife.37072.013

The following source data and figure supplements are available for figure 2:

**Source data 1.** RB-TnSeq analysis of *E. coli's* growth in pairwise conditions on 10% cheese curd agar, pH7.
DOI: https://doi.org/10.7554/eLife.37072.016

**Figure supplement 1.** *E. coli* and community member growth curves alone, in pairwise conditions or during community growth.

*Figure 2 continued on next page*

*Figure 2 continued*

DOI: https://doi.org/10.7554/eLife.37072.014

**Figure supplement 2.** Comparison of the genes important for *E. coli* growth alone, in each pairwise condition or with the community.

DOI: https://doi.org/10.7554/eLife.37072.015

focused on the pairwise-alleviated and pairwise-induced negative fitness as these groups contain genes potentially related to interactions.

Genes whose negative fitness is alleviated by pairwise growth can highlight processes that are of importance for growth alone but no longer important because of the presence of a partner, thus suggesting interactions between *E. coli* and the partner. Just over half of the genes with negative fitness alone appeared to be relieved by the presence of a partner (n = 82 genes, *Figure 2C*), suggesting major modifications of growth conditions following the introduction of a partner. We mapped these 82 alleviated genes to the KEGG BRITE database to identify functions and pathways that are no longer critical in the presence of a partner (*Figure 2E*). 16 genes were associated with unknown or predicted proteins and did not map to any field of the database. Of the remaining genes, 45 mapped to modules of the KEGG orthology hierarchy.

Most of the genes with alleviated negative fitness were associated with the KEGG metabolism module and are thus part of metabolic pathways. It is especially evident that pairwise growth leads to major changes in the need for amino acid biosynthesis. For example, 6 out of the 8 genes of valine and isoleucine biosynthetic pathways are no longer associated with a negative fitness during pairwise growth (*Figure 3C*). In addition, 2 genes of arginine biosynthesis, 2 genes of methionine biosynthesis as well as final steps of homoserine, aspartate and glutamate biosynthesis are no longer required. Moreover, *ilvY*, the transcriptional activator of valine and isoleucine biosynthesis was also among the genes no longer required for pairwise growth. Here, the dominant presence of amino acid biosynthesis genes in the alleviated functions suggests cross-feeding of the pathway end-products or intermediates which are either provided directly by the partner species or made more available in the environment as a consequence of the partner's metabolic activity. Thus, our data suggest that pairwise growth may allow cross-feeding of the amino acids valine, isoleucine, arginine, methionine, homoserine, aspartate and glutamate. Isoleucine and methionine are also intermediates of cofactor biosynthesis, and genes associated with their biosynthesis were also mapped to metabolism of cofactors and vitamins.

To understand if the genes with pairwise-alleviated negative fitness were related to a specific partner, we investigated how each partner contributed to this gene set (*Figure 2—figure supplement 2*). Of the 82 total genes, 36 were alleviated in all pairwise conditions, suggesting that any partner leads to the compensation of these requirements. They included genes associated with amino acid metabolism specific to homoserine and methionine biosynthesis. Of the remaining genes, eight were specifically not required in the presence of *H. alvei*, nine were specifically not required in the presence of *G. candidum* and nine were specifically not required in the presence of *P. camemberti*. Alleviation of leucine and valine biosynthesis was observed with both fungal partners, while biosynthesis of arginine appeared to be no longer required specifically in the presence of *G. candidum*. Fungal species are known to secrete proteases that digest small peptides and proteins (*Kastman et al., 2016*; *Boutrou et al., 2006b*; *Boutrou et al., 2006a*) and may lead to increased availability of amino acids in the environment.

We then analyzed the 74 genes with pairwise-induced negative fitness in order to identify functions or pathways that become important in the presence of a partner (*Figure 2E*). These genes represent almost half (75 out of 153 – *Figure 2C*) of the genes with negative fitness in pairwise conditions, suggesting that presence of a partner introduces new selection pressures. 33 genes mapped to KEGG orthology terms. Among this gene set are pathways associated with signal transduction, biofilm formation and drug resistance. They were related to three major responses: metabolic switch (*creB*: carbon source responsive response regulator), response to stress and toxic compounds (*cpxA*: sensory histidine kinase, *oxyR*: oxidative stress regulator, *acrAB*: multidrug efflux) and biofilm formation (*rcsC* and *rcsB*: regulator of capsular synthesis, *pgaC*: poly-*N*-acetyl-D-glucosamine synthase subunit). Biofilms are microbial structures known to provide resistance to different stresses, including resistance to antibiotics, and biofilm-inducing genes can be activated in the presence of stress events (*Landini, 2009*). The transcriptional regulator OxyR and the transduction

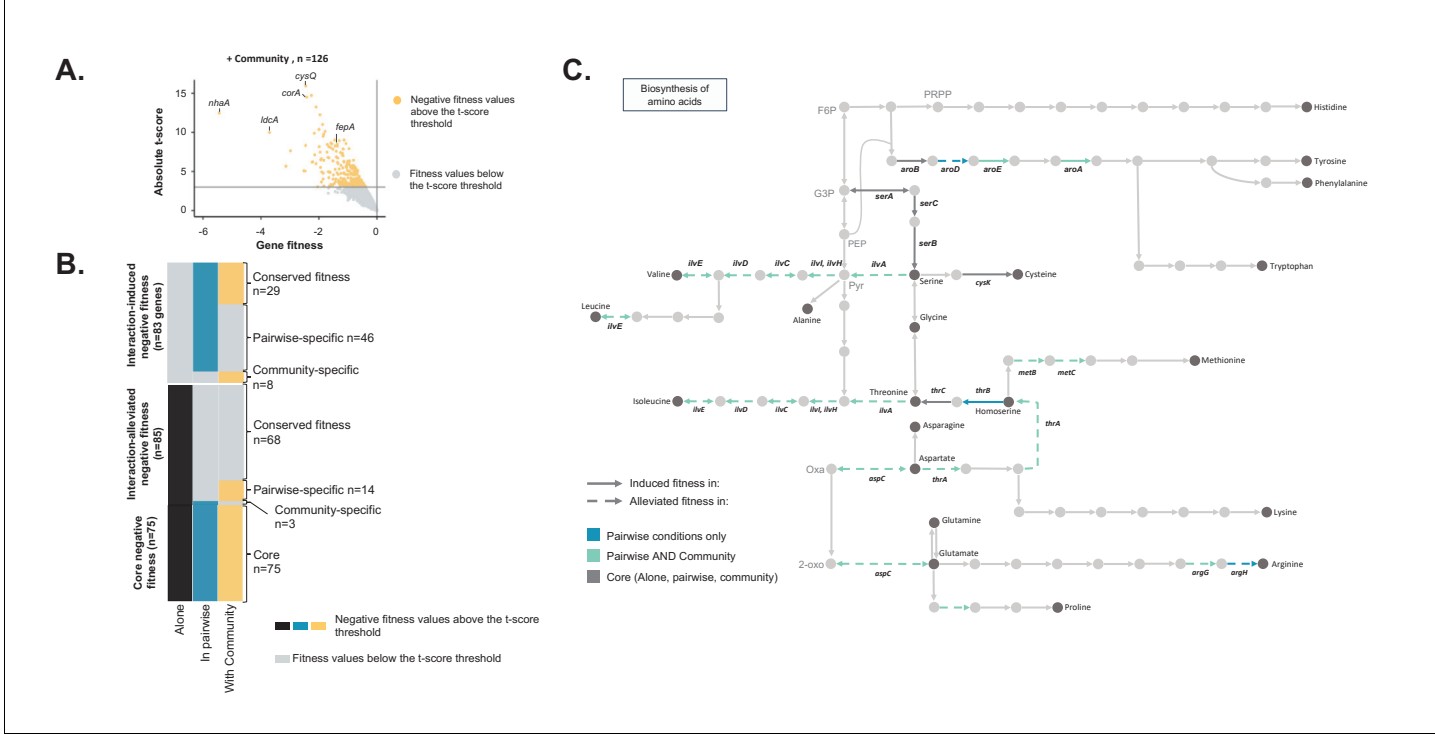

**Figure 3.** Comparison of *E. coli* genes with negative fitness within the community and in pairwise conditions (*Figure 3—source data 1*). (**A**) Using the *E. coli* RB-TnSeq library, we identified genes required to grow with the community (*H. alvei* + *G. candidum* + *P. camemberti*). During growth with the community, we identified a total of 126 genes with a negative fitness. (**B**) We compared the pairwise-induced and community-induced genes (Interaction-induced genes) as well as the pairwise-alleviated and community-alleviated genes (Interaction-alleviated genes) to identify conservation of interactions from pairwise to community and emergence of higher-order interactions. (**C**) Within the alleviated negative fitness, genes associated with numerous amino acid biosynthetic pathways were identified. F6P: fructose-6-phosphate, PRPP: 5-phospho-ribose-1-di-phosphate, G3P: Glyceraldehyde-3-phosphate, PEP: phosphoenol-pyruvate, Pyr: Pyruvate, Oxa: Oxaloacetate, 2-oxo: 2-oxoglutarate.

DOI: https://doi.org/10.7554/eLife.37072.017

The following source data is available for figure 3:

**Source data 1.** RB-TnSeq analysis of *E. coli*'s growth with the community on 10% cheese curd agar, pH7.
DOI: https://doi.org/10.7554/eLife.37072.018

system CpxA and CpxB are known coordinators of stress response and biofilm formation (*Gambino and Cappitelli, 2016*; *Dorel et al., 2006*). While these genes represent only a small subset of the pairwise-induced gene set, they could suggest that partner species are producing toxic compounds or oxidative stress-inducing compounds.

We again investigated if these responses were partner-specific (*Figure 2—figure supplement 2*). Of the 74 pairwise-induced negative fitness, 11 were found to have a negative fitness in the presence of all partners, 13 were specific to the presence of *H. alvei*, 24 were specific to the presence of *G. candidum* and 11 were specific to the presence of *P. camemberti*. Despite involving different genes, necessity of biofilm formation and response to toxic stress were associated with the presence of all partners.

Finally, functional analysis of the conserved genes with negative fitness highlighted that functions associated with membrane transport, including resistance to high osmolarity and iron transport as well as functions associated with energy metabolism and aromatic amino acid biosynthesis were still important to grow in the presence of a partner (*Figure 2E*).

We performed similar pairwise assays using the RB-TnSeq library of *P. psychrophila* JB418 with *H. alvei*, *G. candidum* or *P. camemberti*. We identified a similar number of genes associated with pairwise-alleviated and pairwise-induced requirements (*Figure 1—figure supplement 4*) as we did when using the *E. coli* library. As with *E. coli*, we can infer production of toxic stress by the partners as

genes associated to DNA repair were identified with a negative fitness in pairwise conditions in the functional analysis. However, cross-feeding by fungal partners was not as striking as for *E. coli*.

## Identification of *E. coli* genetic requirements for growth within the community and comparison to pairwise conditions

We next aimed to investigate the differences between genes with a negative fitness during growth in a community (complex interactive condition) and genes with a negative fitness during growth in associated pairwise conditions (simple interactive conditions) (*Figure 2—figure supplement 2*). We grew the *E. coli* library with the complete community composed of *H. alvei*, *G. candidum* and *P. camemberti* and we identified 126 genes with a reliable negative fitness (*Figure 3A*, *Figure 3— source data 1*). *E. coli*'s final biomass was more reduced by the presence of the community than by a single partner. However, the growth of each community member remained unaffected (*Figure 2— figure supplement 1*).

We first identified community-induced and community-alleviated genes by comparing the genes with a negative fitness in the community with the genes with a negative fitness during growth alone. We identified 89 genes that had negative fitness for both community and alone (conserved negative fitness), 37 genes with negative fitness only with the community (community-induced negative fitness) and 71 genes with negative fitness only for growth alone (community-alleviated negative fitness). As with a single partner, the presence of a complex community potentially relieves some fitness effects while introducing new ones.

Comparing community-induced and pairwise-induced genes can reveal if and how community complexity modifies the genes that are important in different interactive contexts compared to growth alone (*Figure 3B* – Interaction-induced negative fitness). We identified 29 genes with a negative fitness in both pairwise and community growth compared to growth alone (conserved interaction-induced negative fitness). These include genes associated with oxidative stress and biofilm formation. These genes are likely to be associated with pairwise interactions which are maintained in a community context.

Meanwhile, eight genes appeared to be specifically associated with negative fitness in the presence of the community (*Figure 3B*, community-specific induced genes), highlighting higher-order interactions that emerge from a higher level of complexity in the community composition. Interestingly, these genes represent only a small fraction (22%) of the community-induced requirements, suggesting that most of the negative fitness effects observed in the community are derived from pairwise interactions.

Finally, we identified 46 genes that have a negative fitness in pairwise conditions, but not during growth alone or within the community (*Figure 3B*, pairwise-specific induced genes). These genes could be related to interactions that are either alleviated or counteracted in a community, either by the presence of a specific species or by the community as a whole. For example, some of the identified genes were associated with antimicrobial resistance, and, in a diverse community, other species could degrade the putative antimicrobial molecules or prevent the producing species from secreting it. Consequently, *E. coli* would be exposed to a lower level of antimicrobials, suppressing the necessity of a resistance gene. Thus, the complex pattern of requirements for these genes may reflect higher-order interactions.

We next investigated if the interactions related to pairwise-alleviated negative fitness and community-alleviated negative fitness were similar (*Figure 3B* – Interaction-alleviated negative fitness). 68 genes were no longer associated with a negative fitness in both pairwise conditions and with the community compared to growth alone (conserved interaction-alleviated negative fitness). These genes may represent pairwise interactions maintained in the community context. Amino acid biosynthesis was highly represented within these genes and more specifically biosynthesis of valine, isoleucine, methionine, homoserine, aspartate and glutamate (*Figure 3C*). This suggests that, despite the presence of more species, these amino acids are still cross-fed.

We also identified 14 genes that no longer had a negative fitness in pairwise conditions compared to growth alone yet remained with a negative fitness in growth with the community (pairwise-specific alleviated negative fitness). These 14 genes represent a small fraction of the pairwise-alleviated, thus suggesting that most of interactions related to pairwise-alleviation are maintained in the community. Finally, only three genes were specifically alleviated in the community (community-specific alleviated fitness). This points out that presence of the full community does not lead to

emergence of specific alleviation of fitness effects but that most of the fitness effect alleviations observed in the community are conserved from pairwise interactions. In both cases, these 14 pairwise-specific and three community-specific alleviated genes could highlight existence of more higher-order interactions. However, too few genes are involved to determine the exact nature of these interactions.

Finally, we identified 75 genes with negative fitness in all conditions (core negative fitness). These genes encompass functions including iron uptake and response to high osmolarity. Overall, they are associated with response to environmental parameters that other species do not alleviate.

To summarize, the community-induced genes were mostly maintained from pairwise-induced genes. Similarly, the genes that were community-alleviated were highly similar to the pairwise-alleviated genes. However, we also observed emergence of higher-order interactions in the community condition as numerous interactions observed in pairwise conditions (n = 46 + 14) were not conserved in the community condition and specific interactions (n = 8 + 3) were observed in the community condition. Altogether, 58% of the interactions observed in the community were from pairwise interactions while 42% emerged from higher community complexity.

Again, we carried out similar experiments and analysis using the *P. psychrophila* JB418 RB-TnSeq library generated in our laboratory. The results were highly similar to the ones observed with *E. coli* in terms of number of genetic requirements alleviated in the presence of the community compared to growth alone as well as the number genes specifically important to grow with the community compared to growth alone (*Figure 1—figure supplement 4*). Finally, we consistently observed importance of higher-order interactions, 61% of the observed interactions in the community were conserved from pairwise interactions and 39% were higher-order interactions.

## Differential expression analysis of *E. coli* in interactive conditions versus growth alone

So far, we used a genome-scale genetic approach to investigate potential microbial interactions. As a complementary strategy, we generated transcriptomic data for *E. coli* during growth in each previously described condition. Changes in transcriptional profiles can be a powerful indicator of an organism's response to an environment and have been used to identify *E. coli* pathways involved in interactions (*Croucher and Thomson, 2010*; *McAdam et al., 2014*; *Galia et al., 2017*).

To measure *E. coli* gene expression, we extracted and sequenced RNA from each timepoint and condition of the same samples used for RB-TnSeq above (after 1, 2 and 3 days of growth when grown alone, in pairwise conditions or with the community). Comparison of transcriptional profiles suggests a strong reorganization of *E. coli* gene expression in response to the presence of a partner (*Figure 4A*, *Figure 4—source data 1* and *Figure 4—figure supplement 1*).

We first focused on the genes differentially expressed between growth in pairwise conditions and growth alone. We calculated the fold change of gene expression between pairwise growth and growth alone and identified differentially expressed genes by screening for adjusted p-values lower than 1% (Benjamini-Hochberg correction for multiple testing) and an absolute log2 of fold change (logFC) greater than 1. To remain consistent with the analysis performed for the genetic requirements, we pooled the data of all timepoints after identifying the upregulated or downregulated genes for each timepoint. We found a total of 966 upregulated and 977 downregulated genes across all partners (482 upregulated genes and 478 downregulated genes in presence of *H. alvei*, 633 upregulated genes and 719 downregulated genes in presence of *G. candidum*, 626 upregulated genes and 694 downregulated genes in presence of *P. camemberti*, *Figure 4A*). Almost half of *E. coli*'s genome is subjected to expression modification, suggesting a global response to the presence of a partner. We further investigated if differential expression in pairwise conditions is partner-specific (*Figure 4—figure supplement 1*). Around half of *E. coli* gene expression regulation in the presence of a partner appears to be independent of which partner is present. Also, a number of genes were differentially expressed depending on the partner: 66 genes were specifically upregulated and 60 genes downregulated with *H. alvei*, 213 upregulated and 182 downregulated with *G. candidum*, and 183 upregulated and 161 downregulated with *P. camemberti*.

Due to the larger gene set compared to RB-TnSeq, we performed KEGG pathway enrichment analyses on the differentially expressed genes in pairwise conditions to determine upregulated functions and pathways (*Figure 4B*). First, almost all of the aminoacyl-tRNA-synthetases and functions associated with energy production were upregulated. Interestingly upregulation of energy

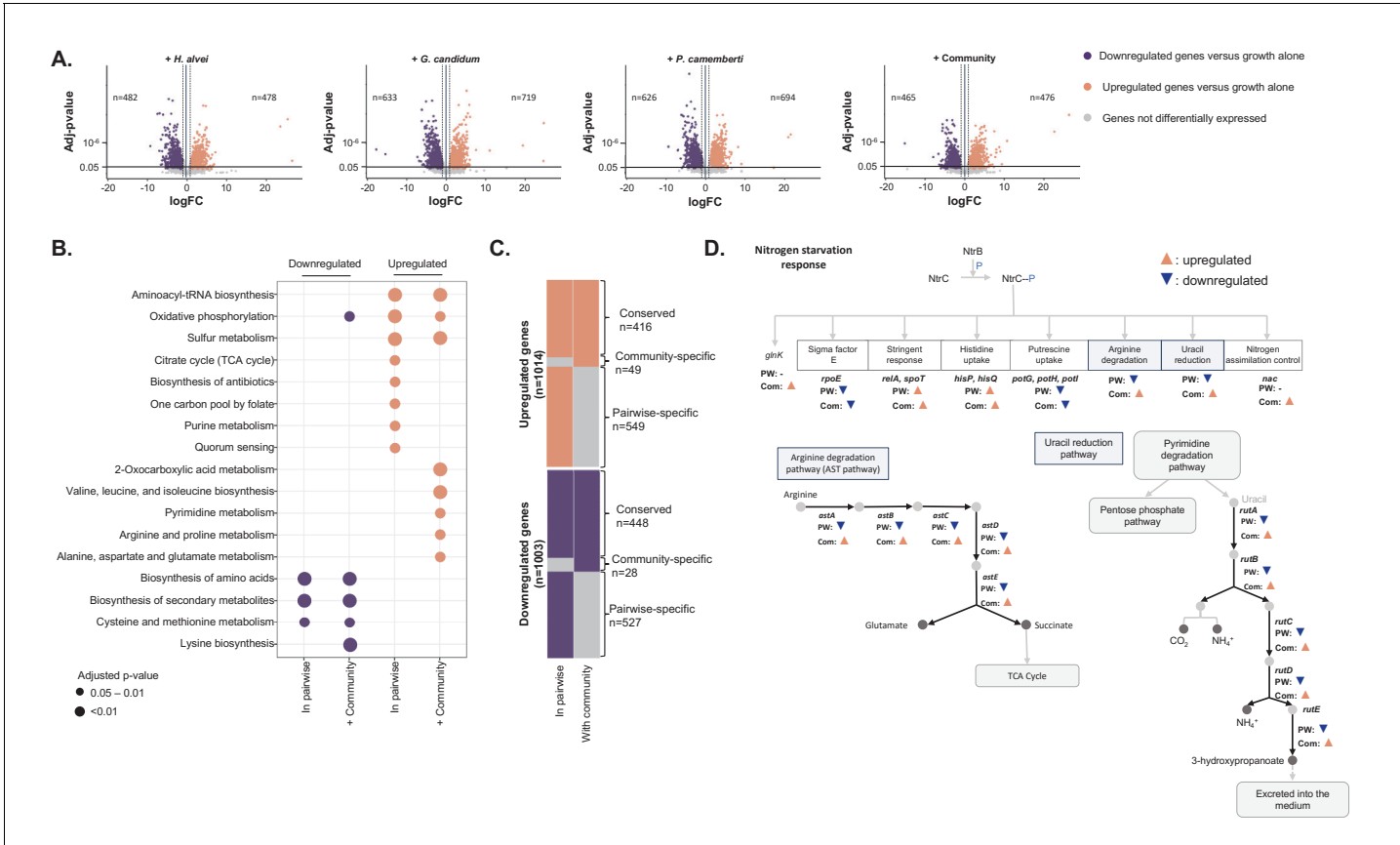

**Figure 4.** Differential expression analysis of *E. coli* during interactive and non-interactive growth conditions (*Figure 4—source data 1*). We used RNASeq to investigate *E. coli* gene expression at three timepoints (1, 2 and 3 days) during growth on CCA alone, in pairwise conditions (with *H. alvei*, *G. candidum* or *P. camemberti*) and with the community. (A) Using DESeq2 (*Love et al., 2015*), we identified up and downregulated genes during growth in each pairwise condition compared to growth alone as well as up and downregulated genes during growth with the community compared to growth alone. Differential expression analysis has been performed at three timepoints, however, we displayed the results of the three timepoints on a single volcano plot. Only genes associated with an adjusted p-value lower than 1% (Benjamini-Hochberg correction for multiple testing) and an absolute logFC higher than one were considered differentially expressed. (B) We regroup any genes upregulated in at least one pairwise condition as a single set of pairwise-upregulated genes and did the same for pairwise-downregulated genes. Then, we performed functional enrichment analysis on KEGG pathways for pairwise-downregulated genes, community-downregulated genes, pairwise-upregulated genes and community-upregulated genes. Functional enrichment was performed using the R package clusterProfiler (*Yu et al., 2012*) and only the KEGG pathways enriched with an adjusted p-value lower than 5% (Benjamini-Hochberg correction for multiple testing) were considered. (C) We compared pairwise-upregulated genes with community-upregulated genes and pairwise-downregulated genes with community-downregulated genes to identify if expression regulation from pairwise conditions is conserved in the community context and if we observe specific changes in pairwise or community conditions. (D) Within the genes specifically upregulated during growth with the community, we observed the upregulation of multiple genes associated with the nitrogen starvation response. Most of these genes were also downregulated in pairwise conditions.

DOI: https://doi.org/10.7554/eLife.37072.019

The following source data and figure supplement are available for figure 4:

**Source data 1.** Differential expression analysis of *E. coli*'s growth in pairwise and with the community versus growth alone.
DOI: https://doi.org/10.7554/eLife.37072.021

**Figure supplement 1.** RNASeq analysis of *E. coli* gene expression during growth alone and in pairwise conditions.
DOI: https://doi.org/10.7554/eLife.37072.020

production through aerobic respiration and the TCA cycle happened after 3 days of growth. Oxygen availability (*Gunsalus, 1992*) and growth phase (*Wackwitz et al., 1999*) are the two known regulators of aerobic respiration. At day 3, *E. coli* was observed to be in log phase when alone, whereas in the presence of a partner, and especially with *P. camemberti*, *E.coli* was observed to enter the stationary phase between day 2 and day 3 (*Figure 2* – figure supplement 1). Therefore, upregulation of aerobic respiration is most likely associated with the growth stage difference between *E. coli* alone

and with a partner. While these functions were upregulated regardless of the partner, more genes were upregulated in the presence of *G. candidum* than the other partners and thus, several pathways associated with nucleotide biosynthesis (C1-pool by folate, purine metabolism, and pyrimidine metabolism) were specifically upregulated with this partner. This suggests that either *E. coli* and *G. candidum* compete for nucleotide compounds from the environment or that presence of *G. candidum* leads to an increased demand of nucleotide compounds for *E. coli's* metabolism and growth.

We performed a similar KEGG pathway enrichment analysis on the downregulated genes in pairwise conditions. Pathways involved in the biosynthesis of amino acids, specifically tyrosine, phenylalanine, tryptophan, methionine, lysine, arginine, homoserine, leucine, glutamate, threonine and glycine, appeared to be the principal downregulated functions in the presence of a partner and more particularly with a fungal partner. Interestingly, some amino acid biosynthetic pathways were upregulated later in the growth but not significantly enriched in the enrichment analysis (phenylalanine, tyrosine and leucine). Downregulation of amino acid biosynthesis suggests that the partner species generates amino acids available for cross-feeding. The observation of this interaction in the transcriptome data is consistent with our interpretation of RB-TnSeq results and reinforces the likelihood of such an interaction. However, late upregulation of some amino acid biosynthesis suggests that as the partner grows along with *E. coli* they eventually end up competing for amino acids, leading to biosynthesis upregulation. This late competition was unlikely to be detected by RB-TnSeq using our current analysis.

To summarize, presence of a partner triggers a significant and dynamic reorganization of *E. coli* gene expression. Most of these modifications restructure *E. coli* metabolic activity: mostly in response to modification of growth phase, but also in response to nutrient availability changes and for example to benefit from cross-feeding and common goods.

Next, we aimed to determine whether *E. coli* gene expression reorganization significantly changes when grown with the full community as compared to growth in pairwise conditions. To do so, we first calculated *E. coli* gene logFC at each timepoint between growth with the community and growth alone. We further analyzed genes with adjusted p-values lower than 1% (Benjamini-Hochberg correction for multiple testing) and absolute logFC greater than 1. After pooling across timepoints, we identified 465 upregulated and 476 downregulated genes in the presence of the community versus growth alone (*Figure 4A*). We then compared these genes to the 966 upregulated genes and 977 downregulated genes in pairwise conditions versus growth alone (*Figure 4B and C*).

First, 416 genes were found to be upregulated in both pairwise and community growth versus growth alone (conserved upregulated genes). Enrichment analysis highlighted functions that were previously described as upregulated in most of the pairwise conditions: aminoacyl-tRNA-synthetase and energy metabolism (*Figure 4B*). This suggests that certain interactions that *E. coli* experienced in pairwise conditions are conserved in the community context. To investigate if the addition of similar interactions from different partners leads to an amplified response, we explored if the magnitude of expression changes in these pathways is higher in the community. We performed differential expression analysis on the genes comparably regulated in pairwise conditions and with the community (*Figure 4—figure supplement 1*). 50 of the 416 conserved upregulated genes were significantly more upregulated in community growth compared to pairwise growth. Among them, sulfate assimilation genes were overrepresented. This suggests that similar pairwise interactions may be additive in the community, leading to a stronger transcriptional response.

Next, we identified 549 genes that were specifically upregulated in pairwise conditions versus growth alone and not upregulated in community versus growth alone (pairwise-specific upregulated genes). KEGG pathway enrichment analysis highlighted that these genes were mostly associated with quorum sensing, fatty acid metabolism and oxidative phosphorylation (*Figure 4B*). This observation suggests that the presence of additional species in the community counteracts or prevents certain pairwise interactions. It supports the presence of higher-order interactions as highlighted with the RB-TnSeq experiments. Indeed, more than half of the upregulated genes observed in pairwise conditions are not conserved with the community.

Finally, 49 genes were specifically upregulated during community growth versus growth alone (community-specific upregulated genes). Emergence of specific expression patterns with the community also suggests the existence of higher-order interactions. However, these community-specific upregulated genes represent only a small fraction (10%) of upregulated genes within the community. Thus, most expression upregulation observed with the community is conserved from expression

upregulation observed in pairwise conditions. Genes specifically upregulated with the community were associated with the biosynthesis of valine, leucine, and isoleucine, pyrimidine metabolism as well as arginine and proline metabolism (*Figure 4B*). Upregulation of certain amino acid biosynthesis pathways suggests that despite potential cross-feeding from individual partners, addition of many partners eventually leads to competition. Upregulation of pyrimidine, arginine and proline metabolism however is part of a larger response; the response to nitrogen starvation (*Figure 4D*). This response facilitates cell survival under nitrogen-limited conditions. Specifically, upregulated genes included all the genes involved in the regulatory loop of the transcriptional regulator NtrC (*glnL*) and nitrogen utilization as well as NtrC transcriptional targets: the transcriptional regulator Nac (*nac*), the operon *rutABCDEFG* involved in ammonium production by uracil catabolism, the *astABCDE* operon constituting the arginine degradation pathway (AST pathway) and the two regulators of the stringent response, *relA* and *spoT*. Thus, the presence of additional species in the community specifically triggers the activation of the response to nitrogen starvation, which suggests a potential higher competition for nitrogen in the community context.

We performed a similar analysis on downregulated genes in pairwise conditions and with the community versus growth alone to investigate if transcriptional downregulation in pairwise and community conditions are similar (*Figure 4C*). We identified 448 genes that were downregulated during both pairwise and community growth conditions versus growth alone (conserved downregulated genes). Enrichment analysis pointed to the downregulation of amino acid biosynthesis as well as cysteine and methionine metabolism. Therefore, consistent with our RB-TnSeq data, this suggests that cross-feeding from a single partner is maintained in a more complex context. 527 genes were specifically downregulated in pairwise conditions and not with the community (pairwise-specific downregulated genes). Despite the large number of genes, no specific functions were enriched. However, the *rutABCDEFG* and *astABCDE* operons associated with the response to nitrogen starvation were downregulated in each pairwise condition (*Figure 4D*). Altogether, pairwise-specific downregulated genes represent 54% of the genes downregulated in pairwise conditions, thus strongly suggesting higher-order interactions. Here, the presence of the community may trigger a highly specific response that would otherwise be downregulated in the presence of only one species. Finally, also highlighting potential higher-order interactions, 28 genes were specifically downregulated when *E. coli* is grown with the community (community-specific downregulated genes). However, this represents only 6% of the observed downregulated genes in the community, highlighting again that most of the gene expression regulations in the presence of the community are conserved from gene expression regulations pairwise interactions.

To conclude, most of the changes in *E. coli* gene expression during growth with the community were similar to a subset of expression changes observed in pairwise conditions. Moreover, some of these changes were amplified in the community compared to pairwise. This suggests that while a large part of transcriptional regulation in the community results from pairwise interactions, similar interactions from different partners may be additive in the community and exert a stronger impact on transcription. Also, the observed changes in nitrogen availability-related transcription suggest that community growth may induce new metabolic limitations.

## Discussion

In this work, we used the model organism *E. coli* as a readout for microbial interactions in a model cheese rind microbiome. We used genome-scale approaches to determine the changes in *E. coli*'s genetic requirements and gene expression profiles in conditions with increasing levels of community complexity. Our analysis highlighted both important changes in *E. coli*'s genetic requirements between interactive and non-interactive conditions as well as deep reorganization of *E. coli*'s gene expression patterns. We identified a variety of interactive mechanisms in the different interactive contexts. Our data revealed that interactions within the community include both competitive and beneficial interactions. By reconstructing a community from the bottom up, we were able to investigate how interactions in a community change as a consequence of being in a more complex, albeit still simple, community. RNASeq and RB-TnSeq consistently showed that around half of the interactions in a community can be attributed to pairwise interactions and the other half can be attributed to higher-order interactions. Although community structure is argued to be predictable from pairwise interactions in specific cases, higher-order interactions are believed to be responsible for the

general lack of predictability (*Billick and Case, 1994*; *Friedman et al., 2017*; *Momeni et al., 2017*). Similarly, such higher-order interactions have been shown to be responsible for the unpredictability of community function from individual species traits (*Sanchez-Gorostiaga et al., 2018*). Our work demonstrates the existence and prevalence of these higher-order interactions even within a simple community.

Together, RB-TnSeq and RNASeq provided insight into mechanisms of mutualism between microbial species in this model system. One major interaction mechanism appears to be cross-feeding of amino acids from fungal partners. Although amino acid biosynthesis pathways were strongly required when *E. coli* grew alone, the presence of fungal species, but not bacterial species, led to fitness effect alleviation and downregulation of amino acid biosynthesis. This suggests that fungi increase the availability of free amino acids in the environment. Cheese-associated fungal species are known to secrete proteases that can degrade casein, the major protein found in cheese (*Kastman et al., 2016*; *Boutrou et al., 2006b*; *Boutrou et al., 2006a*), and therefore may increase the availability of an otherwise limiting resource. Although our model system is based on cheese, interactions based on cross-feeding are widely observed in other environments, such as soil, the ocean or the human gut (*Freilich et al., 2011*; *Pacheco et al., 2018*; *Goldford et al., 2018*). For example, in the gut microbiome, Bifidobacteria can ferment starch and fructooligosaccharides and produce fermentation products including organic acids such as acetate which can in turn be consumed by butyrate-producing bacteria like *Eubacterium hallii* (*Belenguer et al., 2006*; *De Vuyst and Leroy, 2011*; *Flint et al., 2012*). Cross-feeding of other nutrients in the gut has also been uncovered using a related approach (INSeq) which found that vitamin B12 from Firmicutes or Actinobacteria was important for the establishment of *Bacteroides thetaiotaomicron* in mice (*Goodman et al., 2009*).

Our results also revealed mechanisms of competition within the community. RNASeq highlighted that both siderophore production and uptake are upregulated in interactive conditions, suggesting that there is competition for iron between species. Competition for iron is frequently observed across many environments, including cheese, as iron is an essential micronutrient for microbial growth and often a limited resource (*Monnet et al., 2012*; *Albar et al., 2014*; *Stubbendieck and Straight, 2016*; *Traxler et al., 2012*). Interestingly, although we were able to detect fitness defects for siderophore uptake using RB-TnSeq, we did not see fitness defects for siderophore biosynthesis mutants. Because RB-TnSeq relies on a pooled library of mutants, one of the limitations to this approach is that it is difficult to detect fitness effects for genes associated with the production of common goods. For example, in the pooled library, most cells have wild-type siderophore biosynthesis genes, and thus produce and secrete siderophores into the environment under iron limitation. A consequence of this is that any cell that has lost the ability to produce siderophores can readily access the siderophores produced by neighboring cells. In contrast, the genes for uptake of common goods should remain crucial, and accordingly, we do observe fitness defects in the siderophore uptake genes. For this reason, using RNASeq can help overcome some of the limitations, such as pooling effects, associated with RB-TnSeq.

Interactions between species also appeared to lead to stressful growth conditions, as RB-TnSeq showed the need for genes to deal with growth in the presence of toxic compounds. *G. candidum* is known to produce and excrete D-3-phenyllactic acid and D-3-indollactic acid, which inhibit the growth of Gram-negative and Gram-positive bacteria in the cheese environment (*Boutrou and Guéguen, 2005*; *Dieuleveux et al., 1998*). Also, strains of *H. alvei* isolated from meat have been shown to produce compounds inhibiting biofilm formation in *Salmonella enterica* serovar Enteritidis (*Chorianopoulos et al., 2010*). To begin to understand the extent to which the interactions we detected with *E. coli* were specific to this species, or more general, we performed similar RB-TnSeq experiments with the cheese isolate *Pseudomonas psychrophila*. This comparative approach showed that some responses to growth with other species are conserved, such as those needed to survive stress conditions, while others differ between the two species such as amino acid cross-feeding. This further highlights the ability to detect the dynamic nature of interactions, which not only change with community complexity, but also with the composition of the community.

While our analysis highlighted global changes occurring as a consequence of interactions, and some of the key underlying interaction mechanisms, many more aspects of the biology occurring within communities are likely to be uncovered even within this simple model system. For example, much of our current analysis is limited to well-characterized pathways with strong negative fitness

effects, yet many uncharacterized genes were also identified as potentially involved in interactions. Further investigation of these genes could uncover novel interaction pathways. Additionally, analysis of the exact ways in which community members modify the growth environment, such as through the production of extracellular metabolites, will be important to fully understand the molecular mechanisms of interactions.

Altogether, this study revealed the intricacy, redundancy and specificity of the many interactions governing a simple microbial community. The ability of *E. coli* to act as a probe for molecular interactions, the robustness of RB-TnSeq, and its complementarity with RNASeq open new paths for investigating molecular interactions in more complex communities, independently of the genetic tractability of their members, and can contribute to a better understanding of the complexity and diversity of interactions within microbiomes. Finally, our work provides a starting point for better understanding the exact nature of higher-order interactions, and how they impact microbial communities.

# Materials and methods

**Key resources table**

| Reagent type (species) or resource | Designation | Source or reference | Identifiers | Additional information |
|---|---|---|---|---|
| Library, strain background (*Escherichia coli* K12) | Keio collection | PMID: 16738554 | CGSC, RRID:SCR_002303 | Collection of 3,818 *E. coli* knockout strains |
| Library, strain background (*Escherichia coli* K12) | Keio_ML9 | PMID: 25968644 | RB-TnSeq library of *E. coli* K12 BW25113 (152,018 pooled insertion mutants) | |
| Library, strain background (*Pseudomonas psychrophila*) | JB418_ECP1 | this paper | | RB-TnSeq library generated in the *P. psychrophila* JB418 strain isolated from cheese (272,329 pooled insertion mutants) |
| Strain, strain background (*Escherichia coli* K12) | Keio ME9062 | PMID: 16738554 | CGSC#: 7636 | Parent strain of the Keio collection mutants. Also referred as *E. coli* K12 BW25113 |
| Strain, strain background (*Hafnia alvei*) | *Hafnia alvei JB232* | this paper | | Strain isolated from cheese |
| Strain, strain background (*Geotrichum candidum*) | *Geotrichum candidum* | Danisco - CHOOZIT | GEO13 LYO 2D | Industrial starter for cheese production |
| Strain, strain background (*Penicillium camemberti*) | *Penicllium camemberti* | Danisco - CHOOZIT | PC SAM 3 LYO 10D | Industrial starter for cheese production |
| Strain, strain background (*P. psychrophila*) | *Pseudomonas psychrophila JB418* | this paper | | Strain isolated from cheese |
| Strain, strain background (*E. coli*) | *E. coli APA766* | PMID: 25968644 | | donor WM3064 which carries the pKMW7 Tn5 vector library containing 20 bp barcodes |
| Sequence-based reagent | NEBNext Multiplex Oligos for Illumina (Set 1); NEBNext multiplex Oligos for Illumina (Set 2) | New England Biolabs | NEB #E7335 (lot 0091412);, NEB #E7500 (lot 0071412) | |
| Sequence-based reagent | Nspacer_barseq_pHIMAR; P7_MOD_TS_index3 primers | PMID: 25968644 | | Primers for transposon-insertion sites amplication for *P. psychrophila* RB-TnSeq library characterization |
| Sequence-based reagent | BarSeq_P1; BarSeq_P2_ITXXX | PMID: 25968644 | | Primers for RB-TnSeq PCR (amplification of the barcode region of the transposon) |
| Commercial assay or kit | NEBNext Ultra DNA Library Prep Kit for Illumina | New England Biolabs | NEB #E7645 | |
| Commercial assay or kit | MinElute purification kit | Qiagen | ID:28004 | |
| Commercial assay or kit | Turbo DNA-free kit | AMBION, Life Technologies | AM1907 | |

*Continued on next page*

*Continued*

| Reagent type (species) or resource | Designation | Source or reference | Identifiers | Additional information |
|---|---|---|---|---|
| Commercial assay or kit | MEGAclear Kit Purification for Large Scale Transcription Reactions | AMBION, Life Technologies | AM1908 | |
| Commercial assay or kit | Ribo-Zero rRNA removal kit (bacteria); Ribo-Zero rRNA removal kit (yeast) | Illumina | MRZMB126; MRZY1306 | |
| Commercial assay or kit | NEBNextUltraTM RNA Library Prep Kit for Illumina | New England Biolabs | NEB #E7770 | |
| Software, algorithm | Geneious | http://www.geneious.com | | |
| Software, algorithm | Perl | https://www.perl.org/ | | |
| Software, algorithm | R | https://www.r-project.org/ | | |
| Other | MapTnSeq.pl; DesignRandomPool.pl; BarSeqTest.pl | PMID: 25968644 | | Perl scripts for RB-TnSeq library characterization and RB-TnSeq analysis - https://bitbucket.org/berkeleylab/feba |
| Other | DESeq2 | PMID: 25516281 | | R package for RNASeq analysis |

## Strains and media
### Strains
The following strains have been used to reconstruct the bloomy rind cheese community: *H. alvei* JB232 isolated previously from cheese (*Wolfe et al., 2014*) and two industrial cheese strains: *G. candidum* (*Geotrichum candidum* GEO13 LYO 2D, Danisco – CHOOZIT™, Copenhagen, Denmark) and *P. camemberti* (PC SAM 3 LYO 10D, Danisco - CHOOZIT™). The strain *P. psychrophila* JB418 was isolated from a sample of Robiola due latti (Italy) (*Wolfe et al., 2014*) and used for all the experiments involving *Pseudomonas*. All the *E. coli* strains used in this study shared the same genetic background of the initial strain *E. coli* K12 BW25113. The use of the different strains is described in *Table 1*.

### Medium
All growth assays have been carried out on 10% cheese curd agar, pH7 (CCA) (10% freeze-dried Bayley Hazen Blue cheese curd (Jasper Hill Farm, VT), 3% NaCl, 0.5% xanthan gum and 1.7% agar). The pH of the CCA was buffered from 5.5 to 7 using 10M NaOH.

## Growth curve assays on 10% cheese curd agar, pH7
The following growth assays are distinct from the growths carried out for RB-TnSeq and fitness analysis (see below).

Assays have been performed in at least triplicates. Growth assays have been carried out for the *E. coli* JW0024 strain (*Baba et al., 2006*) and *P. psychrophila* JB418 during growth alone, in pairwise conditions with either *H. alvei* JB232, *G. candidum* or *P. camemberti* and with the full community.

*E. coli* was pre-cultured overnight in liquid LB-kanamycin (50 µg/ml) at 37°C and *P. psychrophila* JB418 was pre-cultured overnight in LB at room temperature (RT) . Then, for growth alone assays, 1000 cells of *E. coli* or *P. psychrophila* JB418 were inoculated on a 96 well plate containing 200 µL of

**Table 1.** *E. coli* strains used during the study.

| Experiment | *E. coli* strain(s) | Reference |
|---|---|---|
| RB-TnSeq | *E. coli* Keio_ML9 library | (*Wetmore et al., 2015*) |
| Growth assays | *E. coli* JW0024 strain (undisrupted mutant) | (*Baba et al., 2006*) |
| Competition assays | WT: Keio ME9062 Mutants: (*Figure 1—figure supplement 5*) | (*Baba et al., 2006*) |

DOI: https://doi.org/10.7554/eLife.37072.022

CCA per well. For pairwise growth assays, either *E. coli* or *P. psychrophila* JB418 was co-inoculated with either *H. alvei* JB232, *G. candidum* or *P. camemberti* at a ratio of 1:1 cell (1000 cells of *E. coli* and 1000 cells of the partner). Finally, for growth assay with the community, *E. coli* or *P. psychrophila* JB418 have been co-inoculated with *H. alvei* JB232, *G. candidum* and *P. camemberti* at a ratio of 10:10:10:1 cells.

Growth assays were then carried out for 3 days at RT. Agar plugs from 96 well plates were harvested at T = 0 hr, 6 hr, 12 hr, 24 hr, 36 hr, 48 hr, 72 hr and 120 hr for *E. coli* growth assays and T = 0 hr, 12 hr, 24 hr, 48 hr and 72 hr for *P. psychrophila* JB418 growth assays. Agar plugs were homogenized in 1 mL of PBS1X-Tween0.05% and three dilutions were plated on different media to measure growth of each species (see *Table 2*). Plates were incubated for 24 hr at 37°C for *E. coli* and 2 days at RT for *P. psychrophila* JB418. After incubation, colony forming units (CFUs) were counted to estimate the number of bacterial cells on the cheese curd agar plates.

Growth alone of *H. alvei* JB232, *G. candidum* and *P. camemberti* have also been carried out similarly to *E. coli* and *P. psychrophila* JB418 growth alone.

## *P. psychrophila* JB418 genome sequencing, assembly and annotation

*P. psychrophila* JB418 gDNA was sequenced using Pacific Biosciences (PacBio), Oxford Nanopore Minion (Oxford Nanopore, Oxford, UK) and Illumina sequencing. PacBio library preparation and sequencing were performed by the IGM Genomics Center at the University of California San Diego. Nanopore library preparation and sequencing were done at the University of California, Santa Barbara as part of the Eco-Evolutionary Dynamics in Nature and the Lab (ECOEVO17). Illumina library preparation and sequencing were done at the Harvard University Center for Systems Biology. Canu was used to assemble the PacBio and nanopore reads (*Koren et al., 2017*). Illumina data was then used to correct sequencing error using the software Pilon (*Walker et al., 2014*). The assembled genome was annotated using the Integrated Microbial Genomes and Microbiomes (IMG/M) system (*Markowitz et al., 2012*). The *P. psychrophila* JB418 genome is 6,072,477 nucleotides long. It contains a single circular chromosome of 5.85 Mb and 4 plasmids of 172.2 Kb, 37.7 Kb, 5.8 Kb and 2.4 Kb. 6060 genes including 5788 open reading frames were identified. This genome is publicly available on the IMG/M website as IMG Genome ID 2751185442.

## Transposon mutant library construction in *P. psychrophila* JB418

*P. psychrophila* JB418 was mutagenized by conjugation with *E. coli* strain APA766 (donor WM3064 which carries the pKMW7 Tn5 vector library containing 20 bp barcodes) (*Wetmore et al., 2015*). This donor strain is auxotrophic for diaminopimelic acid (DAP). The full collection of the APA766 donor strain (1 mL) was grown up at 37°C overnight at 200 rpm. Four 25 mL cultures (each started with 250 μL of APA766 stock) were grown in LB-kanamycin:DAP (50 μg/mL kanamycin and 60 μg/mL DAP). A 20 mL culture was started from an individual *P. psychrophila* JB418 colony in LB broth and grown at RT overnight at 200 rpm. *E. coli* donor cells were washed twice with LB and resuspended in 25 mL LB. Donor and recipient cells were then mixed at a 1:1 cell ratio based on OD600 measurements, pelleted, and resuspended in 100 μL. This was done separately for each of the four *E. coli* cultures. 40 μL were plated on nitrocellulose filters on LB plates with 60 μg/mL DAP. Two filters were used for each of the four conjugation mixtures (eight total conjugations). The conjugations took place for 6 hr at RT. After 6 hr, the filters were each resuspended in 2 mL of LB broth and then plated on LB:kanamycin (50 μg/mL) for selection of transconjugants. 20 plates were plated of a 1:2

**Table 2.** Organization of CFU's quantification for growth assays.

| *E. coli* + *H. alvei* JB232 | LB (*E. coli* + *H. alvei* JB232 CFUs) LB-kanamycin (50 μg/ml) (*E. coli* CFUs) |
|---|---|
| *E. coli* + *G. candidum* | LB-kanamycin:cycloheximide (50 μg/ml and 10 μg/ml) (*E. coli* CFUs) LB-chloramphenicol (*G. candidum* CFU's) |
| *E. coli* + *P. camemberti* | LB-kanamycin:cyclohexamide (50 μg/ml and 10 μg/ml) (*E. coli* CFUs) LB-chloramphenicol (50 μg/ml)(*P. camemberti* CFU's) |
| *E. coli* + Community | LB-cyclohexamide (10 μg /mL) (*E. coli* and *H. alvei* JB232 CFU's), LB-kanamycin:cyclohexamide (50 μg/ml and 10 μg/ml) (*E. coli* CFU's) and LB-chloramphenicol (50 μg/ml) (*G. candidum* and *P. camemberti* CFU's) |

DOI: https://doi.org/10.7554/eLife.37072.023

dilution for each conjugation (160 plates total). Transconjugants were pooled and harvested after three days of growth on selection plates. The pooled mixture was diluted back to 0.25 in 100 mL of LB:kanamycin (50 µg/mL). The culture was then grown at RT to an OD600 of 1.3 before glycerol was added to 10% final volume and 1 mL aliquots of the library (named JB418_ECP1) were made and stored at −80°C for future use.

## TnSeq sequencing library preparation for *P. psychrophila* JB418 and TnSeq data analysis

Library preparation was performed as in *Wetmore et al., 2015* with slight modifications (*Wetmore et al., 2015*).

### DNA extraction

DNA was extracted from the *P. psychrophila* JB418_ECP1 RB-TnSeq library by phenol:chloroform extraction. Briefly, the cell pellet was vortexed at maximum speed for 3 min in the presence of 500 µL buffer B (200 mM NaCl,20mM EDTA sterilized by filtration), 210 µL of 20% SDS, a 1:1 mixture of 425–600 µM and 150–212 µm acid-washed beads, and 500 µL of phenol:chloroform, pH 8. The sample was then centrifuged for 3 min at 4°C at 8000 rpm prior to removing the aqueous phase to a new tube. 1/10 of sample aqueous phase volume of 3M sodium acetate was then added along with one aqueous phase volume of ice cold isopropanol. The sample was then placed for ten minutes at −80°C before centrifugation for five minutes at 4°C at 13000 rpm. The supernatant was discarded and 750 µL of ice cold 70% ethanol was added before another centrifugation for five minutes at 4°C at 13000 rpm. The supernatant was discarded and the DNA pellet was allowed to air dry before resuspension in 50 µL of nuclease-free water. DNA was quantified with Qubit double-stranded DNA high-sensitivity assay kit (Invitrogen, Carlsbad, CA).

### DNA fragmentation and size selection

2 µg of DNA was sheared with a Covaris E220 focused-ultrasonicator with the following settings: 10% duty cycle, intensity 5, 200 cycles per burst, 150 s. DNA was split into two aliquots (1 µg each) and samples were size-selected for fragments of 300 bp using 0.85X Agencourt AMPure XP beads (Invitrogen) with a 1.4x ratio following the manufacturer's instructions.

### Library preparation

The entire 20 µL volume of these two size-selected samples were then each used as input into the NEBNext End Prep step 1.1 of the NEBNext Ultra DNA Library Prep Kit for Illumina (New England Biolabs, Ipswich, MA) protocol. The remainder of the manufacturer's protocol was then followed with the exception that for adapter ligation, we used 0.8 µL of 15 µM double-stranded Y adapters. Adapters were prepared by first combining 5 µL of 100 µM Mod2_TS_Univ (ACGCTCTTCCGATC*T) and 5 µL of 100 µM Mod2_TruSeq (/5'P/GATCGGAAGAGCACACGTCTGAACTCCAGTCA. This mixture was then incubated in a thermocycler for 30 min at 37°C, followed by ramping at 0.5°C per second to 97.5°C before a hold at 97.5°C for 155 s. The temperature was then decreased by 0.1°C per five seconds for 775 cycles, followed by a hold at 4°C. Annealed adapters were diluted to 15 µM in TE and stored at −80°C before use. AMPure XP ratios for a 200 bp insert size were used as recommended in *Table 1.1* of the NEBNext Ultra DNA Library Prep Kit for Illumina manual.

To enrich for transposon-insertion sites, PCR amplification was done on the adapter-ligated DNA with NEBNext Q5 Hot Start HiFi Master Mix and Nspacer_barseq_pHIMAR and P7_MOD_TS_index3 primers (*Wetmore et al., 2015*) with the following program: 98°C 30 s, 98°C 10 s, 65°C 75 s, repeat steps 2–3 24X, 65°C 5 min, and then maintained at 4°C. Following PCR and clean-up of step 1.5 of the NEBNext Ultra DNA Library Prep Kit for Illumina manual, the two preps were pooled and the concentration was quantified with Qubit double-stranded DNA high-sensitivity assay kit (Invitrogen). A second size selection clean-up was performed by repeating step 1.5 of the NEBNext Ultra DNA Library Prep Kit for Illumina manual.

## Library sequencing

The sample was analyzed on an Agilent TapeStation and the average size was 380 bp and the concentration was 57 pg/μL. This sample was then submitted for sequencing on a HiSeq 2500 Rapid Run (150 bp fragments, paired-end) at the UCSD IGM Genomics Center.

## Library characterization

TnSeq reads were analyzed with the Perl script MapTnSeq.pl from (*Wetmore et al., 2015*). This script maps each read to the *P. psychrophila* genome. The script DesignRandomPool.pl (*Wetmore et al., 2015*) was used to generate the file containing the list of barcodes that consistently map to a unique location as well as their location. We obtained a total of 272,329 insertion mutants. The transposon was inserted in the central part of a gene for 143,491 of these insertion mutants covering 83% of *P. psychrophila* JB418 genome (*Figure 1—figure supplement 6*).

# RB-TnSeq experiments for *E. coli* and *P. psychrophila* JB418

The *E. coli* barcoded transposon library Keio_ML9 and the *P. psychrophila* strain JB418 library were used for RB-TnSeq fitness assays on CCA during growth alone, growth in pairwise condition with each bloomy rind cheese community member and during growth with the full community. *Figure 1—figure supplement 1* provides a description of the fitness assays as well as fitness calculation.

## Library pre-culture

Each library has to be initially amplified before use. One aliquot of each library was thawed and inoculated into 25 mL of liquid LB-kanamycin (50 μg/mL). Once the culture reached mid-log phase (OD = 0.6–0.8), 5 mL of that pre-culture was pelleted and stored at −80°C for the T0 reference in the fitness analysis. The remaining cells were used to inoculate the different fitness assay conditions.

## Inoculations

For each RB-TnSeq fitness assay, $7*10^6$ cells of the library pre-culture were inoculated by spreading evenly on a 100 mm petri dish containing 10% CCA, pH seven after having been washed in PBS1x-Tween0.05%. This represents on average 50 cells per insertion mutant. For each pairwise assay, $7*10^6$ cells of the partner were co-inoculated with the library. For the community assay, $7*10^6$ cells of *H. alvei* JB232 and *G. candidum* as well as $7*10^5$ cells of *P. camemberti* were co-inoculated with the library. For each condition, assays were performed in triplicate.

## Harvest

Harvests were performed at T = 24 hr, 48 hr and 72 hr. Sampling was done by flooding a plate with 1.5 mL of PBS1X-Tween0.05% and gently scraping the cells off. The liquid was then transferred into a 1.5 mL microfuge tube and cells were pelleted by centrifugation. After removing the supernatant, the cells were washed in 1 mL of RNA-protect solution (Qiagen, Hilden, Germany), pelleted and stored at −80°C before further experiments.

## gDNA and mRNA extraction

gDNA and mRNA were simultaneously extracted by a phenol-chloroform extraction (pH 8) from samples of the competitive assays. For each extraction: 125 μL of 425–600 μm acid-washed beads and 125 μL of 150–212 μm acid-washed beads were poured in a screw-caped 2 mL tube. 500 μL of 2X buffer B (200 mM NaCl, 20 mM EDTA) and 210 μL of SDS 20% were added to the tube as well as the pellet and 500 μL of Phenol:Chloroform (pH 8). Cells were lysed by vortexing the tubes for 2 min at maximum speed. Aqueous and organic phases were separated by centrifugation at 4°C, 8,000 RPM for 3 min and 450 μL of the aqueous phase (upper phase) was recovered in a 1.5 mL eppendorf tube. 45 μL of sodium acetate 3M and 450 μL of ice cold isopropanol were added before incubating the tubes at −80°C for 10 min. The tubes were then centrifuged for 5 min at 4°C at 13,000 RPM. The pellet was then washed in 750 μL of 70% ice cold ethanol and re-suspended in 50 μL of DNAse/RNAse free water. Each sample was split into 2 times 25 μL and stored at −80°C until further analysis.

## Library preparation and sequencing

After gDNA extraction, the 98°C BarSeq PCR as described in *Wetmore et al., 2015* was used to amplify only the barcoded region of the transposons. Briefly, PCR was performed in a final volume of 50 μL: 25 μL of Q5 polymerase master mix (New England Biolab), 10 μL of GC enhancer buffer (New England Biolab), 2.5 μL of the common reverse primer (BarSeq_P1 – *Wetmore et al., 2015*) at 10 μM, 2.5 μL of a forward primer from the 96 forward primers (BarSeq_P2_ITXXX) at 10 μM and 50 ng to 2 μg of gDNA. For each triplicate, the PCR was performed with the same forward primer so all replicates of a condition could be pooled and have the same sequencing multiplexing index. For *E. coli* analysis, we performed 46 PCRs (T0 sample and 45 harvest samples) involving 16 different multiplexing indexes. For *P. psychrophila* JB418 analysis, we performed 46 PCR (T0 sample and 45 harvest samples) involving 16 other multiplexing indexes. We used the following PCR program: (i) 98°C - 4 min, (ii) 30 cycles of: 98°C – 30 s; 55°C – 30 s; 72°C – 30 s, (iii) 72°C – 5 min. After the PCR, 10 μL of each of the 92 PCR products were pooled together to create the BarSeq library(920 μL) and 200 μL of the pooled library were purified using the MinElute purification kit (Qiagen). The final elution of the BarSeq library was performed in 30 μL in DNAse and RNAse free water.

The BarSeq library was then quantified using Qubit dsDNA HS assay kit (Invitrogen) and sequenced on HiSeq4000 (50 bp, single-end reads), by the IGM Genomics Center at the University of California San Diego. The sequencing depth for each condition varied between 1.5 and 7.5 million reads.

## Data processing and fitness analysis

BarSeq data processing and gene fitness calculation were performed separately for the *E. coli* and the *P. psychrophila* JB418 experiments. For each library, BarSeq reads were processed using the Perl script BarSeqTest.pl from (*Wetmore et al., 2015*). This script combines two Perl scripts essential for the BarSeq data processing. After the raw reads have been de-multiplexed, the computational pipeline: (i) identifies individual barcodes and the associated number of reads, (ii) calculates the strain fitness for each insertion mutant and (iii) calculates the normalized fitness value for each gene along with a t-statistic value (t-score). The following parameters were applied during the fitness calculations: (i) only insertion mutants located within the central region of genes (10%–90%) were considered, (ii) barcodes with less than three reads in the T0 were ignored and (iii) genes with less than 30 counts across all barcodes in T0 were ignored. For each library, the pipeline uses a table where each barcode is mapped to a location in the genome. The Arkin lab (Physical Biosciences Division, Lawrence Berkeley National Laboratory, Berkeley, California, USA) kindly provided the TnSeq table for the *E. coli* library and we generated a TnSeq table for *P. psychrophila* strain JB418. The different scripts used for this analysis originate from (*Wetmore et al., 2015*) and are publicly available on https://bitbucket.org/berkeleylab/feba.

We calculated *E. coli* and *P. psychrophila* JB418 genes fitnesses at T = 24 hr (Day1), 48 hr (Day2) and 72 hr (Day3) in the following conditions: growth alone, growth with *H. alvei*, growth with *G. candidum*, growth with *P. camemberti* and growth with the community.

First, strain fitness for each insertion mutant that met the criteria described above is calculated as the log2 of the ratio of the insertion mutant's abundance at the time of the harvest (number of reads of the associated barcode) and its abundance in the T0 sample. Un-normalized gene fitness is then calculated as the weighted average of strain fitness of all the insertion mutants of a gene. Un-normalized fitness values are then normalized, first by subtracting the smoothed median of the un-normalized fitness values. This is performed to account for changes in gene copy number along the chromosome as genes close to the replication fork might have multiple copies in diving cells. Then, the final normalization step relies on the assumption that disruption of most of the genes leads to little to no fitness effect. This normalization is performed by subtracting the mode of the gene fitness. Thus, most of the genes are expected to have a fitness of 0. Genes whose disruption is deleterious will have a negative fitness and genes whose disruption is beneficial a positive fitness. A t-score is calculated along with each gene fitness to evaluate how reliably different from zero the gene fitness is. The t-score is a moderated t-statistic calculated as the ratio of the gene fitness and its standard deviation. More details can be found in *Wetmore et al., 2015*.

In this study, all our experiments and genes fitness values met the quality requirements to be further analyzed (*Figure 1—figure supplement 7*).

## Keio collection mutant competition assays for RB-TnSeq validation

We used mutants from the Keio collection to validate the genes identified by RB-TnSeq as having a significant fitness in *E. coli* growth alone on CCA (see list in *Figure 1—figure supplement 5*). Each mutant was grown in a competition assay with the non-kanamycin resistant wild-type (Keio ME9062 –(*Baba et al., 2006*)). 1000 cells of a specific mutant were inoculated with 1000 cells of the wild type (WT) on the surface of the same cheese plug in a 96 well plate containing 10% CCA, pH7. The number of the mutant cells and the WT cells were calculated at T0 and day one after harvesting and homogenizing the cheese plug, plating serial dilutions and counting CFUs. Experimental fitness of each mutant was calculated as the log2 of the ratio of the mutant abundance (mutant CFUs divided by total CFUs (WT +mutant)) after 24 hr and its abundance at T0.

## RNASeq and differential expression analysis

### RNASeq library preparation

Libraries were prepared in duplicate for the following conditions: *E. coli* growth alone, with *H. alvei*, with *G. candidum*, with *P. camemberti* and with the community for T = 24 hr, 48 hr and 72 hr. RNA samples from the *E. coli* BarSeq experiment were used to produce the RNASeq library.

Each library was prepared as follows. First, RNA samples were treated with DNAse using the 'Rigorous DNAse treatment' for the Turbo DNA-free kit (AMBION, Life Technologies, Waltham, MA) and RNA concentration was measured by nucleic acid quantification in Epoch Microplate Spectrophotometer (BioTek, Winooski, VT). Then, transfer RNAs and 5S RNA were removed using the MEGAclear Kit Purification for Large Scale Transcription Reactions (AMBION, Life Technologies) following manufacturer instructions. Absence of tRNA and 5S RNA was verified by running 100 ng of RNA on a 1.5% agarose gel and RNA concentration was quantified by nucleic acid quantification in Epoch Microplate Spectrophotometer. Also, presence of trace amounts of genomic DNA was assessed by PCR using universal bacterial 16S PCR primers (Forward primer: AGAGTTTGATCCTGGCTCAG, Reverse Primer: GGTTACCTTGTTACGACTT). The PCR was performed in a final volume of 20 µL: 10 µL of Q5 polymerase master mix (New England Biolabs), 0.5 µL of forward primer 10 uM, 0.5 µL of reverse primer 10 uM and 5 µL of non-diluted RNA. PCR products were run on a 1.7% agarose gel and if genomic DNA was amplified, another DNAse treatment was performed as well as a new verification of absence of genomic DNA. Ribosomal RNA depletion was performed using the Ribo-Zero rRNA removal kit by Illumina (Illumina, San Diego, CA). According to manufacturer instructions; we used 1 µL of RiboGuard RNAse Inhibitor in each sample as suggested and followed instructions for 1–2.5 ug of RNA input and we used a 2:1 mix of bacterial Ribo-Zero Removal solution and yeast Ribo-Zero Removal solution. rRNA depleted samples were recovered in 10 µL after ethanol precipitation. Concentrations after ribodepletion were measured using Qubit RNA HS Assay Kits (Invitrogen). The RNASeq library was produced using the NEBNextUltraTM RNA Library Prep Kit for Illumina for purified mRNA or ribosome depleted RNA. We prepared a library with fragments size of 300 nucleotides and used the 10 uM NEBNext Multiplex Oligos for Illumina (Set 1, NEB #E7335) lot 0091412 and the NEBNext multiplex Oligos for Illumina (Set 2, NEB #E7500) lot 0071412. We performed PCR product purification with 0.8X Agencourt AMPure XP Beads instead of 0.9X. Library samples were quantified with Qubit DNA HS Assay Kits before the quality and fragment size were validated by TapeStation (HiSensD1000 ScreenTape). Library samples were pooled at a concentration of 15 nM each and were sequenced on HiSeq4000 (50 bp, single-end).

### Differential expression analysis

RNASeq reads were mapped to the concatenated genome of *Escherichia coli* K12 BW25113 (*Grenier et al., 2014*) and *H. alvei* using Geneious version R9 9.1.3 (http://www.geneious.com, [*Kearse et al., 2012*]). Only the reads that uniquely mapped to a single location on the *E. coli* genome section were conserved. *E. coli* and *H. alvei* genome are divergent enough so 50 nucleotide reads potentially originating from *H. alvei* mRNA would not map to the *E. coli* genome and few reads from *E. coli* would map on the *H. alvei* genome.

*E. coli* expression analysis was performed using the following R packages: Rsamtool (R package version 1.30.0), GenomeInfoDb (R package version 1.14.0.), GenomicFeatures (*Lawrence et al., 2013*), GenomicAlignments, GenomicRanges (*Lawrence et al., 2013*) and DESeq2 (*Love et al., 2015*). We followed the workflow described by Love et al. and performed the differential expression

analysis using the package DESeq2. Differentially expressed genes between two conditions were selected with an adjusted p-value lower than 1% (Benjamini-Hochberg correction for multiple testing) and an absolute log2 of fold change equal to or greater than 1.

## KEGG pathway enrichment analysis

Functional enrichment analysis was performed using the R package clusterProfiler (*Yu et al., 2012*). We used the latest version of the package org.EcK12.eg.db for *E. coli* annotations (R package version 3.5.0.). We used Benjamini-Hochberg for multiple comparison correction and only the KEGG pathways enriched with an adjusted p-value lower than 5% were considered.

## Acknowledgments

The authors would like to thank: the Arkin lab and the Deutschbauer lab at UC-Berkeley for the *E. coli* Keio_ML9 library and their help for the RB-TnSeq analysis, Steven Villareal and Tyler Nelson for their help in the competitive assays, Kristen Jepsen at the IGM Genomics Center at the University of California San Diego for assistance with sequencing, Ben Wolfe and Sandeep Venkataram for their constructive comments on this work as well as all the Dutton lab members for their input. Nanopore sequencing of *P. psychrophila* JB418 was supported in part by NSF Grant No. PHY17-48958, NIH Grant No. R25GM067110, and the Gordon and Betty Moore Foundation Grant No. 2919.01

## Additional information

### Funding

| Funder | Grant reference number | Author |
|---|---|---|
| CJS INRA/INRIA | | Manon Morin |
| National Institutes of Health | 5 T32 GM 7240-40 | Emily C Pierce |
| David and Lucile Packard Foundation | #2016-65131 | Rachel J Dutton |
| Pew Charitable Trusts | Pew Scholar in Biomedical Sciences | Rachel J Dutton |
| National Institutes of Health | P50 GM068763 | Rachel J Dutton |

The funders had no role in study design, data collection and interpretation, or the decision to submit the work for publication.

### Author contributions

Manon Morin, Conceptualization, Data curation, Software, Formal analysis, Validation, Investigation, Visualization, Methodology, Writing—original draft, Writing—review and editing; Emily C Pierce, Data curation, Software, Formal analysis, Investigation, Visualization, Writing—review and editing; Rachel J Dutton, Conceptualization, Resources, Data curation, Supervision, Funding acquisition, Writing—original draft, Project administration, Writing—review and editing

### Author ORCIDs

Manon Morin (iD) http://orcid.org/0000-0003-2158-0473
Emily C Pierce (iD) http://orcid.org/0000-0002-9960-0270
Rachel J Dutton (iD) http://orcid.org/0000-0002-2944-2182

### Decision letter and Author response

Decision letter https://doi.org/10.7554/eLife.37072.031
Author response https://doi.org/10.7554/eLife.37072.032

# Additional files

## Supplementary files

• Supplementary file 1. Differential expression analysis of *E. coli* growth in community versus pairwise conditions. We used RNASeq to investigate changes in *E. coli* gene expression between growth in the community and in each pairwise conditions (with *H. alvei*, *G. candidum* or *P. camemberti*). Using DESeq2 (*Love et al., 2015*), we identified up and downregulated genes during growth in community versus growth in each pairwise condition individually. Only genes associated with an adjusted p-value lower than 1% (Benjamini-Hochberg correction for multiple testing) and an absolute log of fold change higher than one were considered differentially expressed.

DOI: https://doi.org/10.7554/eLife.37072.024

• Transparent reporting form

DOI: https://doi.org/10.7554/eLife.37072.025

## Data availability

The Pseudomonas psychrophila JB418 genome is publicly available at https://img.jgi.doe.gov/ (IMG Genome ID 2751185442).

The following dataset was generated:

| Author(s) | Year | Dataset title | Dataset URL | Database, license, and accessibility information |
|---|---|---|---|---|
| Rachel J Dutton, Emily C Pierce | 2017 | Pseudomonas psychrophila JB418 | https://img.jgi.doe.gov/cgi-bin/m/main.cgi?section=TaxonDetail&page=taxonDetail&taxon_oid=2751185442 | Publicly available at the IMG website (Genome ID 2751185442) |

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
