## [Decision Letter]

Thank you for submitting your article "Changes in the genetic requirements for microbial interactions with increasing community complexity" for consideration by *eLife*. Your article has been reviewed by three peer reviewers, and the evaluation has been overseen by a Reviewing Editor and Wendy Garrett as the Senior Editor. The following individual involved in review of your submission has agreed to reveal his identity: Alvaro Sanchez (Reviewer #3).

The reviewers have discussed the reviews with one another and the Reviewing Editor has drafted this decision to help you prepare a revised submission.

Summary:

In "Changes in the genetic requirements for microbial interaction with increasing community complexity" the authors show that bacteria require a different set of genes to grow alone, in pairwise competition, or in communities. Moreover, they also show that the gene expression profile in the bacteria changes upon the bacterial interaction partner. The addressed question and the conclusion that genetic requirements of bacteria are interaction dependent are very, very interesting.

Please note we are aware of the messaging exchange that clarified a key methodological concern of the reviewers and enabled this request for resubmission. Please make sure that the clarification of these concerns is clearly addressed in the revised submission.

Major concerns:

The mutant library consists of 150,000+ strains. Were all of these strains grown as 150,000 separate colonies on the CCA agar plates? Or were they a complex mixture? If they were grown as a mixture, then different mutants can interact within the mix (e.g. cross-feeding) obscuring the findings of genes relevant for growth on CCA medium. Please clarify the growth conditions and discuss the limitations of the chosen condition.

Spatial aspects:

The experiments were done in a spatial setting, by plating the bacteria on agar. However, this will lead to strong spatial heterogeneity, with areas where only specific mutants grow, whereas interaction may happen only at the borders of those patches. This may obscure a lot of interaction. Moreover, the population expands in a spatial environment, which is known to cause genetic drift that can obscure the fitness outcomes of those bacteria.

Single species cultures were done on 100mm (?) petri dishes but pairwise and community experiments were done in 96-well plates and thus different spatial settings. Why was the experimental design changed between the different experiments? Moreover the number of bacteria per area was much higher in the petri dishes than in the 96-well plates. Since the size of the inoculum will change the spatial organization (fewer cells lead to bigger, homogeneous patches) and the interaction depends on the spatial organization, the change of experimental design from one experiment to the other may alter the outcome and makes me wonder how comparable the experiments are.

Terminology:

All reviewers had trouble with the terminology: "essential" vs. "important" genes. An "essential gene" is one essential for growth. That is not the usage by the authors. Instead, here, "essential genes" are just genes with reliably negative fitness effects upon knock-out. However, this may be true for many (likely even the majority of) genes?

The authors state that the fitness was calculated for 3289 genes. 40 were removed by showing positive fitness effects upon knock out, but in the end only 160 genes were identified as having negative fitness effects. Does it mean that the majority of genes (around 3000) were excluded because they showed non-reliable data (small t-score)? Are the obtained 160 genes not so much 'essential genes' but simply genes that lead to reproducible data? In other words, the majority of gene-knockout could lead to decreased fitness but just some show reliable results and those are called 'essential genes'. How strong is variation for different insertion localizations within the same gene? How sensitively are fitness measures depending on the parameters chosen for the pipeline? How strong is variation between triplicates? Does this explain the many unreliable fitness values?

Timing:

This issue is related to the library pool method that the authors use to determine fitness. Is it that they are pooling together the entire transposon deletion library, throwing it together into their CCA medium, and measuring the fitness at three different time points? What they are putting together is already a community of *E. coli* strains which, collectively, strongly modify their environment as they grow. Thus, this *E. coli* community already contains multiple interactions among mutant strains, which are probably less strong early on during the growth period (say at day 1) when the environmental effects of the community's growth (i.e. through the collective depletion and secretion of resources) would be expected to still be weak-ish. However, on days 2, 3 there should be significant environmental construction by the community, and indeed this is apparent, not only by the fact that the population is not growing anymore after ~40hr (Figure 1A), but by the fact that the genes that show a negative fitness effect on days 2-3 are not the same as those that have an effect on day 1 (Figure 1B).

The authors do have a control for this, by looking at individual mutants that had been scored as having a negative effect within the community and directly competing them against the wild-type after one day of co-culture. In most cases this works well (See Figure 1—figure supplement 5), but they did not do the same for days 2-3, when interactions among deletion mutants that are pooled together may be stronger. A different but related complication to interpret deleterious mutations during days 2-3 (when the multi-strain population is not growing anymore) is the possibility of cell death. The authors measure population size by CFUs, and the population size does barely change after 40hrs. While the obvious explanation is that cells are just not dividing anymore but is it possible that some cell death is occurring but it is balanced by cell divisions? A breakdown of which genes are deleterious on days 1, 2, 3 in each of the conditions (monoculture, pairwise, community) would be helpful to disentangle which genes are deleterious in the unperturbed CCD environment that the authors provide (and whose composition is well understood) versus in the environment constructed by the *E. coli* community, which is much less well understood in this case. This is important in order to adequately rationalize and interpret interactions with other species either in pairwise or multi-species communities; The data presented in Figure 2-3 is also, evidently, pooling together genes that are deleterious in days 1, 2, 3.

An analysis where the genes found to be essential on days 1, 2, 3 are separated (which to some extent they do in Figure 1B, but then they do not incorporate into their analysis any further, as far as I could see) would be worthwhile, and a discussion of the limitations of the method particularly in the context of environmental/niche construction would significantly strengthen the paper.

Discussion points:

Either in the Introduction, or in the Discussion sections, the authors should relate their findings of suggested mechanisms for interactions in communities to related findings from other microbial communities, beyond cheese. In addition, *E. coli* (which the work mainly focuses on, i.e. Figures1-5) does not normally live on cheese rinds and is therefore not a natural interaction partner for the other species. The authors address this issue in the Introduction only very briefly, by pointing out that their transposon screen allows them to identify the genetic requirements for *E. coli* interacting with the other species. This is correct of course, but identifying the relevant interactions for *E. coli* to live with species it never encounters in the real world may be irrelevant. The authors can use the same data and analysis to identify how the 3 cheese-native species interact with other species using *E. coli* as a readout. This kind of analysis is already present in the results description, but a clearer distinction between which side of the interaction is being investigated throughout the manuscript would conceptually strengthen the manuscript. In addition, a clearer and longer description (beyond technological justification) in the Introduction for why using *E. coli* is relevant is needed.

higher-order interactions, subsection “Identification of *E. coli* genes essential for growth within the community and comparison with genes essential for growth in pairwise conditions”, last paragraph: The authors mention that their findings from the TnSeq screen "underline the presence of higher-order interactions". An explanatory paragraph on the relevant genes is necessary. There is currently only one paragraph (see seventh paragraph of the aforementioned subsection), which does not describe anything about these genes except that there are 14+3 relevant genes. Also: this small number of relevant genes seems to be at odds with the rather broad statement of the relevance of higher-order interactions given at the end of the Introduction.

The function of every gene doesn't need to be discussed in the main text (e.g. subsection “Identification of genes essential for *E. coli* growth in pairwise conditions”, fifth paragraph). The *P. psychrophila* story might just be put to the supplement, since it does not really add new things. Either in the Introduction, or in the Discussion sections, the authors should relate their findings of suggested mechanisms for interactions in communities to related findings from other microbial communities, beyond cheese.

Many of the figures show number of genes that are essential for this or that condition. These numbers are hard to interpret. They are not really put into context (e.g. what does the 26 in Figure 2B top really convey? How would it change the overall message if this number were 10 or 50?). In most cases they confuse more than they transport real information. More of a summary of the meaning of the numbers would perhaps be helpful.

Specific questions:

Subsection “Identification of the basic genetic requirements for growth of the *E. coli* sensor species in isolation”, first paragraph: This equation for the fitness cannot be correct, as it will always be positive. Or should the normalization be included in the fitness equation? But which strain has no growth rate defect on CCA medium? The WT? Please clarify and reword. It seems like some mutants had a growth-increase on CCA medium (but were excluded from the analysis).

Subsection “Identification of the basic genetic requirements for growth of the *E. coli* sensor species in isolation”, second paragraph: Why is the number of genes for which the fitness was calculated smaller than the number of genes covered in the transposon library?

Subsection “Identification of the basic genetic requirements for growth of the *E. coli* sensor species in isolation”, last paragraph: Why were only 25 defined mutants tested and not all (since the Keio collection is comprehensive)? How were these 25 mutants chosen?

How many genes did the authors remove based on the criteria of T0? Was there any bias in the genes you removed?

How was fitness value estimated and t-score? assume it is described in Wetmore at al., 2015, but it would be helpful to shortly explain here as well.

How was number of generations measured in Figure 1A? Could the authors show the fitness 'raw data' of the pairwise and community conditions like they showed them for *E. coli* alone in Figure 1B?

---

## [Author Response]

Major concerns:The mutant library consists of 150,000+ strains. Were all of these strains grown as 150,000 separate colonies on the CCA agar plates? Or were they a complex mixture? If they were grown as a mixture, then different mutants can interact within the mix (e.g. cross-feeding) obscuring the findings of genes relevant for growth on CCA medium. Please clarify the growth conditions and discuss the limitations of the chosen condition.

The library is composed of 152018 insertions mutants pooled together. We apologize for any confusion, and have clarified that the library is a mix of mutants growing together in the text (Results – subsection “Identification of the basic genetic requirements for growth of *E. coli* in isolation”). As a mixture, the mutants have the potential to interact with each other, such as through cross-feeding. We agree that this is an important limitation of using a pooled library, and in the original manuscript, we briefly discussed this possibility. We have now added a more detailed discussion of the concerns associated with “pooling” effects.

In the Results: “In our experiment, the fact that all of the mutants are pooled together limits our ability to identify genes whose phenotypes can be complemented by common goods (molecules released in the environment) produced by neighboring cells. […] However, no genes from the enterobactin biosynthesis pathway (*entCEBAH, entD* and *entF*) had a significant negative fitness value (average fitness of the enterobactin biosynthesis pathway: 0.1), while individual growth of these enterobactin biosynthesis mutants from the KEIO collection grew poorly on CCA compared to a rich, non-iron-limited medium (Figure 2—figure supplement 1).”

In the Discussion: “Because RB-TnSeq relies on a pooled library of mutants, one of the limitations to this approach is that it is difficult to detect fitness effects for genes associated with the production of common goods. […] For this reason, using RNASeq can help overcome some of the limitations, such as pooling effects, associated with RB-TnSeq.”

Because we have a very clear instance of how a pooled library will miss genes whose products can be complemented by the products of neighboring cells (common goods), we decided to use this as an example to illustrate this limitation. In Figure 2—figure supplement 1, we show that mutants in the enterobactin siderophore production pathway, when grown individually, fail to grow on the iron-limited cheese curd agar (CCA). However, in our RB-TNseq analysis, these mutants do not show a growth defect as a result of the pooling of mutants in the library. In contrast, our RNAseq analysis shows that these genes are upregulated on CCA, suggesting that siderophore production is important for growth on CCA, consistent with the grow defects we observe of mutants grown individually.

Spatial aspects:The experiments were done in a spatial setting, by plating the bacteria on agar. However, this will lead to strong spatial heterogeneity, with areas where only specific mutants grow, whereas interaction may happen only at the borders of those patches. This may obscure a lot of interaction. Moreover, the population expands in a spatial environment, which is known to cause genetic drift that can obscure the fitness outcomes of those bacteria.

We agree that spatial structure can, and certainly does, impact interactions in an environment that is not well-mixed. However, in our experiment, there are many insertion mutants representing each gene (median of 16 insertion mutants/gene), and we expect these mutants to be randomly distributed on the plate. As a result, all of the mutants of a gene won’t be clustered together in a specific area of the agar plate. Thus we expect that stochastic events (such as potential genetic drift) may randomly affect individual insertion mutants, but are very unlikely to affect all the insertion mutants of a gene. We have addressed that concern in the manuscript.

“As the library is composed of multiple insertion mutants for a gene, we expect the individual insertion mutants to be evenly distributed in the experimental environment, minimizing the effect on any individual insertion mutant due to stochastic processes such as genetic drift or localized effects related to spatial structure (Hallatschek et al., 2007).”

Further, in the RNASeq analysis, the effects of spatial structure should be averaged across all cells. Therefore, in the event that we miss potential interactions with the RB-TnSeq due to spatial structure, we should be able to identify these in RNAseq.

Single species cultures were done on 100mm (?) petri dishes but pairwise and community experiments were done in 96-well plates and thus different spatial settings. Why was the experimental design changed between the different experiments? Moreover the number of bacteria per area was much higher in the petri dishes than in the 96-well plates. Since the size of the inoculum will change the spatial organization (fewer cells lead to bigger, homogeneous patches) and the interaction depends on the spatial organization, the change of experimental design from one experiment to the other may alter the outcome and makes me wonder how comparable the experiments are.

The experiments performed in 96 well plates containing cheese curd agar (Figure 2A) included an *E. coli* “alone” condition, in addition to the pairwise and community conditions. Thus the growth curve experiments are indeed comparable. We apologize for omitting this information from the Materials and methods. We added the appropriate modifications in the Materials and methods section:

“Growth assays have been carried out for the *E. coli* JW0024 strain (Baba et al., 2006) and *P. psychrophila* JB418 during growth alone, in pairwise conditions with either *H. alvei* JB232, *G. candidum* or *P. camemberti* and with the full community. […] Finally, for growth assay with the community, *E. coli* or *P. psychrophila* JB418 have been co-inoculated with *H. alvei* JB232, *G. candidum* and *P. camemberti* at a ratio of 10:10:10:1 cells.”

To avoid further confusion we removed the growth curve associated with growth alone in a petri dish from Figure 1A. The main purpose of the figure was to display the times of sampling and is not crucial for understanding the experiment.

Terminology:All reviewers had trouble with the terminology: "essential" vs. "important" genes. An "essential gene" is one essential for growth. That is not the usage by the authors. Instead, here, "essential genes" are just genes with reliably negative fitness effects upon knock-out. However, this may be true for many (likely even the majority of) genes?

We understand the confusion due to use of the word “essential”. To avoid any confusion, modifications have been made in the text to remove “essential”.

The authors state that the fitness was calculated for 3289 genes. 40 were removed by showing positive fitness effects upon knock out, but in the end only 160 genes were identified as having negative fitness effects. Does it mean that the majority of genes (around 3000) were excluded because they showed non-reliable data (small t-score)?

The reviewers concerns here are mostly due to a confusing explanation of the t-score and an inappropriate use of the word “unreliable”. We hope the following explanation and figures will eliminate this confusion. Similar explanations have also been added in the new manuscript in the Results and Materials and methods sections.

Along with the normalized gene fitness, a t-score is calculated as a moderated t-statistic (see specific details on how the t-score is estimated below). By definition, the t-score values are conceptually similar to a z-score and ideally follow a normal distribution. The t-score is meant to indicate how reliably a gene fitness value is different from 0. In our data analysis, we set-up a t-score threshold of absolute(t-score)>=3, to identify genes with fitness values confidently different from 0 with a false discovery rate of 0.2%. Thus, we aim to identify genes associated with strong fitness defects and filter out genes with subtle to no fitness effect. Moreover, genes fitness normalization relies on the assumption that disruption of most of the genes should lead to subtle to no fitness modification. Consequently, most genes are expected to not have a strong fitness value and to be filtered out by the t-score.

As expected, for each analyzed condition, the vast majority of fitness values (on average 97%) did not pass our t-score threshold and were thus not considered strong fitness effects (Figure 1A). While the majority of the rejected fitness values are associated with low to no fitness effect (fitness value close to 0), some of the rejected fitness values (7%) might appear as strong fitness effects but did not meet our confidence threshold. This group of genes may not pass the confidence threshold due to high variability resulting from: (i) noisy data due to low number of reads in the T0 sample, (ii) a low number of insertion mutants for a gene or (iii) variability within the insertion mutant fitness of a gene associated with insertion location.

Plots presented in Author response image 1 compare these parameters for the genes with significant fitness values and genes with high fitness values (fitness values >0.5 and <-0.5) but not passing the t-score threshold at day 3 during growth alone. The number of insertion mutants and number of reads in the T0 samples appeared to be lower for gene fitness values that did not pass the t-score cutoff than for the genes with significant fitness values. However, we did not observe strong differences associated with variation due to insertion location. Thus, low numbers of insertion mutants within a given gene and/or low sequencing coverage in the T0 sample may increase variability and lower confidence in gene fitness calculation. Higher sequencing depth would decrease the variability attributed to low counts in the T0 sample.

**Author response image 1. respfig1:** Investigation of parameters potentially influencing variability in gene fitness calculation. Here, we compare (i) the number of insertion mutants/gene, (ii) the number of counts/gene in the T0 sample and (iii) the variability associated with insertion location, for genes with fitness values that pass the t-score threshold and genes with strong fitness values (fitness values >0.5 or <-0.5) that don’t pass the t-score threshold.

While variation associated with insertion location did not appear as a general reason for variability, it may it some specific cases be the origin of variation as illustrated in Author response image 2. To account for some of this variability, the analysis pipeline excludes data from insertion mutants located in the beginning and end of genes (first and last 10%).

**Author response image 2. respfig2:** Insertion location and variability of gene fitness values. Upper-part: Quality parameters generated along with gene fitness calculation indicate low variability in gene fitness values associated with insertion location. Lower-part: Examples of insertion mutants’ fitness as a function of the insertion location in the gene showing low (plot1) and high (plot2) variability due to insertion location.

To summarize, the majority of the gene fitness values are filtered out not because they are unreliable but because they have low to no fitness effect. As we aim to find genes with strong fitness effects (absolute t-score >=3), we are not concerned with the genes that are filtered out of the analysis.

Are the obtained 160 genes not so much 'essential genes' but simply genes that lead to reproducible data? In other words, the majority of gene-knockout could lead to decreased fitness but just some show reliable results and those are called 'essential genes'.

As mentioned above, the majority of gene-knockouts had a fitness close to 0, meaning their disruption did not lead to statistically significant defect compared to the other mutants in the library. The 160 genes are the genes whose disruption leads to a growth defect compared to the rest of the library in the studied condition.

How strong is variation for different insertion localizations within the same gene?

As described in Wetmore et al. 2015, different parameters are calculated in the fitness calculation pipeline which allow assessment of the quality of the experiment (i.e. the quality of the calculated fitness). One of these parameters is mad12, defined as the absolute median of difference between fitness in the first half and fitness in the second half of the gene. Another parameter, cor12, calculates the correlation between the fitness in the first half of the gene and the second half of a gene. To ensure that there is not strong dependency of gene fitness on the insertion location, mad12 has to be lower than 0.5 and cor12>= 0.1. These criteria were met in all our experiments (Author response image 2) and were added as Figure 4—figure supplement 1.

Further, for individual genes, insertions within the first and last 10% of a gene are not included in the analysis to reduce bias due to insertion location. For most of the genes, low variation due to insertion location is expected. However, in some specific cases, variation within the insertion mutant fitness values of a same gene can be observed. As an example of potential variation depending on the insertion location, we present below the insertion mutants’ fitness as a function of the insertion location for a gene with low variation of insertion mutant fitness values (*htrC* – growth alone – day 3) and a gene with variation of insertion mutant fitness values associated with insertion location (*pcnB* – growth alone – day 3). We observe a low variance among the insertion mutant fitness of the gene htrC (upper plot) as fitness values are randomly distributed around 0. On the other hand, *pcnB* (lower plot) has a fitness value not close to 0 but not confidently different from 0 according to our confidence criteria. Variation of insertion mutant fitness values depending on the insertion location in this gene may in this specific example be the reason for a low t-score value.

The variation based on insertion location is taken into account in the t-score calculation (see below for details on how the t-score is calculated) as the estimated variance.

How sensitively are fitness measures depending on the parameters chosen for the pipeline?

In the pipeline, parameters affecting gene fitness measures include: (i) exclusion of insertion mutants and genes based on the number of reads in the T0 sample and (ii) exclusion of insertion mutants if the insertion is within the first and last 10% of the gene. These parameters help reduce the noise due to low abundance in the inoculated library or to insertion location. For our analysis, we used the default parameters established by Wetmore et al. as lowering them would increase the noise and variability of data.

How strong is variation between triplicates? Does this explain the many unreliable fitness values?

In our experimental setup, the replicates were pooled for sequencing. qPCR analysis of pairwise and community conditions highlighted that *E. coli* DNA concentration in the DNA extractions was low, presumably due to the presence of DNA from the other species. This would result in only 2ng of *E. coli* DNA instead of the 50ng input recommend for the PCR. Consequently, barcodes with low representation in the library because of strong fitness defects may be missed by the PCR. We decided to pool PCR products of replicates of a condition to increase our chances of detecting barcodes associated with low abundance. Thus we only have one fitness value per condition and cannot assess the variation between replicates within these data. A previous RB-TnSeq analysis of *E. coli* growth alone on CCA in similar inoculation conditions were performed in 4 replicates in the lab and highlighted high correlation between replicates (Pearson coefficient of 0.8). Also, Wetmore et al., 2015 highlighted a correlation between 2 biological replicates ranging from 0.79 to 0.97. Thus, this highlights low variability and high reproducibility of RB-TnSeq data.

As described previously, most of the gene fitness values were not unreliable but close to 0.

Edits have been made in the revised manuscript to remove the confusion about unreliable fitness.

Timing:This issue is related to the library pool method that the authors use to determine fitness. Is it that they are pooling together the entire transposon deletion library, throwing it together into their CCA medium, and measuring the fitness at three different time points? What they are putting together is already a community of E. coli strains which, collectively, strongly modify their environment as they grow. Thus, this E. coli community already contains multiple interactions among mutant strains, which are probably less strong early on during the growth period (say at day 1) when the environmental effects of the community's growth (i.e. through the collective depletion and secretion of resources) would be expected to still be weak-ish. However, on days 2, 3 there should be significant environmental construction by the community, and indeed this is apparent, not only by the fact that the population is not growing anymore after ~40hr (Figure 1A), but by the fact that the genes that show a negative fitness effect on days 2-3 are not the same as those that have an effect on day 1 (Figure 1B).The authors do have a control for this, by looking at individual mutants that had been scored as having a negative effect within the community and directly competing them against the wild-type after one day of co-culture. In most cases this works well (See Figure 1—figure supplement 5), but they did not do the same for days 2-3, when interactions among deletion mutants that are pooled together may be stronger. A different but related complication to interpret deleterious mutations during days 2-3 (when the multi-strain population is not growing anymore) is the possibility of cell death. The authors measure population size by CFUs, and the population size does barely change after 40hrs. While the obvious explanation is that cells are just not dividing anymore but is it possible that some cell death is occurring but it is balanced by cell divisions? A breakdown of which genes are deleterious on days 1, 2, 3 in each of the conditions (monoculture, pairwise, community) would be helpful to disentangle which genes are deleterious in the unperturbed CCD environment that the authors provide (and whose composition is well understood) versus in the environment constructed by the E. coli community, which is much less well understood in this case. This is important in order to adequately rationalize and interpret interactions with other species either in pairwise or multi-species communities; The data presented in Figure 2-3 is also, evidently, pooling together genes that are deleterious in days 1, 2, 3.An analysis where the genes found to be essential on days 1, 2, 3 are separated (which to some extent they do in Figure 1B, but then they do not incorporate into their analysis any further, as far as I could see) would be worthwhile, and a discussion of the limitations of the method particularly in the context of environmental/niche construction would significantly strengthen the paper.

Indeed the transposon library is a pool of 152,018 insertion mutants. Over time, we expect the environment to change in response to the library growth as cells will utilize available nutrients and release molecules, thereby modifying the environment. These abiotic modifications are expected to lead to different genetic requirements at different stages of the growth. For this reason, we decided to calculate gene fitness at multiple timepoints, to encompass as many of the genetic requirements that could be important for any moment of the growth as possible. As the library is pooled and we expect all the insertion mutants (on average 16 per gene) to be evenly distributed on the plate, we expect that the whole library is likely to modify the environment as a wild-type culture would, and thus interactions among insertion mutants won’t lead to specific modifications of the environment. Moreover, the reviewers assume strong interactions between mutants and significant pooling effect within the library. The fact that most of the genes have a fitness value close to 0 suggests that this is not the case.

Although we agree completely that the dynamics of gene fitness requirements over time would be a very interesting future direction, our primary goal in this work was to identify the genes that are important for growth in non-interactive (alone) and interactive conditions (pairwise and community),. As described above, our motivation for collecting data at several timepoints, was to maximize our detection of genes that are important for growth, regardless of timepoint or growth stage. Following gene fitness calculations, we group all the genes that are important at each timepoint as a single group of important genes for each condition. To more accurately reflect this approach, in the revised manuscript, the data generated across timepoints are displayed together as a single dataset per condition.

In order to provide a quantitative description of the differences between timepoints, additional comparative methods are needed. We are currently working on developing these methods, but we believe that this is outside the scope of the current manuscript.

Discussion points:Either in the Introduction, or in the Discussion sections, the authors should relate their findings of suggested mechanisms for interactions in communities to related findings from other microbial communities, beyond cheese.

Such comparisons have been added within the Discussion section in the revised manuscript. We have specifically mentioned examples of cross-feeding in the human gut microbiome.

“Although our model system is based on cheese, interactions based on cross-feeding are widely observed in other environments, such as soil, the ocean or the human gut (Freilich et al., 2011; Pacheco et al., 2018; Goldford et al., 2017). […] Cross-feeding of other nutrients in the gut has also been uncovered using a related approach (INSeq) which found that vitamin B12 from Firmicutes or Actinobacteria was important for the establishment of *Bacteroides thetaiotaomicron* in mice (Goodman et al., 2009).”

In addition, E. coli (which the work mainly focuses on, i.e. Figures1-5) does not normally live on cheese rinds and is therefore not a natural interaction partner for the other species. The authors address this issue in the Introduction only very briefly, by pointing out that their transposon screen allows them to identify the genetic requirements for E. coli interacting with the other species. This is correct of course, but identifying the relevant interactions for E. coli to live with species it never encounters in the real world may be irrelevant. The authors can use the same data and analysis to identify how the 3 cheese-native species interact with other species using E. coli as a readout. This kind of analysis is already present in the results description, but a clearer distinction between which side of the interaction is being investigated throughout the manuscript would conceptually strengthen the manuscript. In addition, a clearer and longer description (beyond technological justification) in the Introduction for why using E. coli is relevant is needed.

The primary rationale for using *E. coli* K12 in this work was due to it being a well-characterized model organism, which would allow us to more easily interpret the biological processes associated with fitness effects. However, other strains of *E. coli* can be found in both raw milk and cheese, including shiga-toxin producing strains contaminating cheese, which have led to foodborne illnesses. We have added this to the Introduction.

“Although *E. coli* K12 is not a typical endogenous species of this particular microbiome, non-pathogenic *E. coli* strains can be found in raw milk and raw-milk cheese (Trmčić et al., 2016). Shiga-toxin-producing *E. coli* 0157:H7 and non-0157 pathogenic *E. coli* species are common invaders of the cheese environment and can survive during cheese making causing mild to life-threatening symptoms after ingestion (Coia et al., 2001; Montet et al., 2009; Frank et al., 1977).”

In addition, clarifications have been added in the introduction and in the discussion concerning whether using *E. coli* as a probe to identify the interactions within the 3 members community or to identify interactions between the community members and *E. coli*.

In the Introduction: “To investigate the genetic basis of microbial interactions, we have adapted this approach to identify and compare genetic requirements in single-species (non-interactive) and multi-species (interactive) conditions. […] The fact that the *E. coli* genome has undergone extensive characterization can help more efficiently interpret the genetic requirements introduced by growth within communities.”

In the Discussion: “In this work, we used the model organism *E. coli* as a readout for microbial interactions in a model cheese rind microbiome. We used genome-scale approaches to determine the changes in *E. coli’s* genetic requirements and gene expression profiles in conditions with increasing levels of community complexity.”

In addition, we use a RB-TnSeq library made in the cheese-isolate species *Pseudomonas psychrophila* as a way to compare the effects we see between this and *E. coli*. We found that these species respond in similar ways in some cases, such as the need for stress resistance pathways, but differ in others, for example in amino acid cross-feeding. The differences could be due to changes in community function in response to *Pseudomonas*, or due to intrinsic differences in the genetic capabilities of the two species. Further work would be needed to elucidate the exact reasons for differences between the genetic requirements observed with *E. coli* and *Pseudomonas*.

higher-order interactions, subsection “Identification of E. coli genes essential for growth within the community and comparison with genes essential for growth in pairwise conditions”, last paragraph: The authors mention that their findings from the TnSeq screen "underline the presence of higher-order interactions". An explanatory paragraph on the relevant genes is necessary. There is currently only one paragraph (see seventh paragraph of the aforementioned subsection), which does not describe anything about these genes except that there are 14+3 relevant genes. Also: this small number of relevant genes seems to be at odds with the rather broad statement of the relevance of higher-order interactions given at the end of the Introduction.

Clarifications have been added in the revised manuscript in the Results subsection “Identification of *E. coli* genetic requirements for growth within the community and comparison to pairwise conditions”. We agree that the higher-order interactions were not highlighted enough. In fact, there were not only 14 genes with negative fitness only with the community + 3 genes with negative fitness alleviated only by the community, but also an additional 46 genes with negative fitness specific to pairwise conditions (alleviated in the community) + 8 genes with negative fitness alleviated by pairwise conditions but not alleviated in the community. Altogether, genes related to higher-order interactions represent around 40% of the observed interactions in the community condition.

The function of every gene doesn't need to be discussed in the main text (e.g. subsection “Identification of genes essential for E. coli growth in pairwise conditions”, fifth paragraph). The P. psychrophila story might just be put to the supplement, since it does not really add new things. Either in the Introduction, or in the Discussion sections, the authors should relate their findings of suggested mechanisms for interactions in communities to related findings from other microbial communities, beyond cheese.

Edits have been made to reduce the function description of individual genes, and the *P. psychrophila* experiments have been shortened in the text and added as supplement as suggested.

Many of the figures show number of genes that are essential for this or that condition. These numbers are hard to interpret. They are not really put into context (e.g. what does the 26 in Figure 2B top really convey? How would it change the overall message if this number were 10 or 50?). In most cases they confuse more than they transport real information. More of a summary of the meaning of the numbers would perhaps be helpful.

We thank the reviewers for pointing this out. We have made extensive changes to the figures to improve clarity. We have also made edits in the text to highlight what information the many numbers convey.

Specific questions:Subsection “Identification of the basic genetic requirements for growth of the E. coli sensor species in isolation”, first paragraph: This equation for the fitness cannot be correct, as it will always be positive. Or should the normalization be included in the fitness equation? But which strain has no growth rate defect on CCA medium? The WT? Please clarify and reword. It seems like some mutants had a growth-increase on CCA medium (but were excluded from the analysis).

The equation described in the former manuscript calculates the un-normalized gene fitness as the weighted average of the strain fitness (i.e. insertion mutant fitness) and is calculated as follows:

Strainfitness=fs=log2nafter+φ-log2(n0+1φ

with nafter and n0 the number of reads of the mutant’s barcode (mutants abundance) at the time of the harvest and in the T0 respectively and φa “pseudocount” to avoid noisy data when nafter is very low to null (as done previously in Wetmore et al., 2015). The strain fitness values are negative whenever nafteris lower than n0. Along with the strain fitness, the strain variance is calculated as:

Strainvariance=Vs=11+nafter+11+n0In(2)2

Then, the unnormalized gene fitness is calculated as follows:

Unnormalized gene fitness=fu=∑i(wifsi)∑iwi

with i representing each insertion mutant of a given gene and wi the weight associated to the strain fitness inversely proportional to its variance.

Then, gene fitness values are normalized. First, a normalization is performed to account for changes in copy number along chromosome (as gene close to the replication fork are expected to have higher copy in dividing cells). This normalization uses the smoothed median within each scaffold (there is only one scaffold in the case of *E. coli*). Then, gene fitness is also normalized so that the typical gene has a fitness of zero. Here, this normalization step does not rely on the WT but on the assumption that disruption of most of the genes leads to subtle to no effects on fitness. Thus, the normalization is performed by subtracting the mode of the gene fitness values distribution from the gene fitness values. Therefore, negative fitness identifies genes whose disruption leads to a growth defect compared to the overall library population and positive fitness identifies genes whose disruption leads to a better fitness than most of the insertion mutants in the library. These methods are described in detail in Wetmore et al., 2015.

We added this information in the revised manuscript (Results and Materials and methods) as well as in a supplementary figure (Figure 1—figure supplement 1) which represents the pipeline of the RB-TnSeq experiments and analysis.

As mentioned in the manuscript, even if we identified positive fitness values, we excluded them from our analysis as we only focused on negative fitness values.

Subsection “Identification of the basic genetic requirements for growth of the E. coli sensor species in isolation”, second paragraph: Why is the number of genes for which the fitness was calculated smaller than the number of genes covered in the transposon library?

The number of genes covered in the library indicates how many *E. coli* genes have at least one insertion mutant with an insertion in the central part of the gene (within the central 10% – 90% of the genes). However, for the analysis, some insertion mutants and genes are removed based on abundance criteria in the T0 sample. To be considered, an insertion mutant must have more than 3 reads in the T0 sample., After removing barcodes with fewer than 3 reads, the total number of reads for all barcodes in a gene are summed, and any gene with fewer than 30 total reads is also removed from the analysis. For these reasons, we end up calculating the fitness for fewer genes than the number of genes covered in the library.

Subsection “Identification of the basic genetic requirements for growth of the E. coli sensor species in isolation”, last paragraph: Why were only 25 defined mutants tested and not all (since the Keio collection is comprehensive)? How were these 25 mutants chosen?

Testing all the 160 mutants in competitive experiment represent a laborious task as it is not an automated process (requires harvest, crushing, plating and then counting CFUs) and validating these 25 mutants took two people a month.

These genes were selected as the ones associated with the strongest fitness defect after one day of growth and what appeared to us as the most interesting functions. Also, we focused on the strongest fitness as CFUs counting might not be adequate to distinguish low but real fitness differences.

How many genes did the authors remove based on the criteria of T0? Was there any bias in the genes you removed?

The library covers 3728 *E. coli* genes and we calculated the fitness of 3298 genes. Thus, 430 genes were removed from that analysis based on the T0 criteria. We did not observe bias in the genes removed in terms of function, genome location or strand location.

How was fitness value estimated and t-score? assume it is described in Wetmore at al., 2015, but it would be helpful to shortly explain here as well.

Please refer to the previous question for fitness value calculation. We added this information in the revised manuscript (Results – subsection “Identification of the basic genetic requirements for growth of *E. coli* in isolation” and Materials and methods) as well as in supplementary figures (Figure 1—figure supplement 1).

The t-score is used to estimate the reliability of fitness of a gene and how reliably it is different from 0. It is calculated using a moderated t statistic, and can be described as the gene fitness divided by the maximum of two estimates of the gene fitness standard deviation:

t=fgσ2+max⁡(Vn,Vo)

Where fg is the normalized gene fitness, σ a small constant (equal to 0.1) to represent uncertainty in the normalization of small fitness values and V is the maximum of two variances: the naïve variance (V_n_) calculated based on the best-case Poisson-noise and number of reads for the gene, and the estimated variance (V_e_) based on the consistency of fitness for the insertion mutant of that gene and particularly adapted for genes with few insertion mutants as it incorporates data from other genes, too. Such information has been added in the Materials and methods of the revised manuscript and in Figure 1—figure supplement 1.

How was number of generations measured in Figure 1A? Could the authors show the fitness 'raw data' of the pairwise and community conditions like they showed them for E. coli alone in Figure 1B?

In the original manuscript, the purpose of Figure 1A was to display when sampling was performed for RB-TnSeq experiments. As this information is not essential for the understanding of the experiment, and further because the number of generations was not used in any part of the fitness calculations, this figure has been removed in the revised manuscript to avoid any confusion. Scatter plots showing the fitness values and the t-scores have been added in Figure 2B and Figure 3A for the different pairwise conditions and for the growth community. Also, the raw data have been added in the figures source files.